# Neuroprotective Potentials of Flavonoids: Experimental Studies and Mechanisms of Action

**DOI:** 10.3390/antiox12020280

**Published:** 2023-01-27

**Authors:** Paolo Bellavite

**Affiliations:** Independent Researcher, 37128 Verona, Italy; paolo.bellavite@gmail.com

**Keywords:** neurotoxicity, neuroinflammation, reactive oxygen derivatives, depression, Parkinson, Alzheimer, flavonoids, hesperidin, quercetin

## Abstract

Neurological and neurodegenerative diseases, particularly those related to aging, are on the rise, but drug therapies are rarely curative. Functional disorders and the organic degeneration of nervous tissue often have complex causes, in which phenomena of oxidative stress, inflammation and cytotoxicity are intertwined. For these reasons, the search for natural substances that can slow down or counteract these pathologies has increased rapidly over the last two decades. In this paper, studies on the neuroprotective effects of flavonoids (especially the two most widely used, hesperidin and quercetin) on animal models of depression, neurotoxicity, Alzheimer’s disease (AD) and Parkinson’s disease are reviewed. The literature on these topics amounts to a few hundred publications on in vitro and in vivo models (notably in rodents) and provides us with a very detailed picture of the action mechanisms and targets of these substances. These include the decrease in enzymes that produce reactive oxygen and ferroptosis, the inhibition of mono-amine oxidases, the stimulation of the Nrf2/ARE system, the induction of brain-derived neurotrophic factor production and, in the case of AD, the prevention of amyloid-beta aggregation. The inhibition of neuroinflammatory processes has been documented as a decrease in cytokine formation (mainly TNF-alpha and IL-1beta) by microglia and astrocytes, by modulating a number of regulatory proteins such as Nf-kB and NLRP3/inflammasome. Although clinical trials on humans are still scarce, preclinical studies allow us to consider hesperidin, quercetin, and other flavonoids as very interesting and safe dietary molecules to be further investigated as complementary treatments in order to prevent neurodegenerative diseases or to moderate their deleterious effects.

## 1. Introduction

Neurological and neurodegenerative disorders have emerged as a major health problem in the current era. One common disorder is nervous depression, while serious diseases, at an organic level too, are Alzheimer’s disease (AD), Parkinson’s disease (PD), Huntington’s disease, multiple sclerosis and amyotrophic lateral sclerosis. These ailments, linked to aging but also to disorders of the metabolic and immune defense systems, lead to the progressive damage of the neurons, often accompanied by the accumulation of pathological substances, chronic inflammation and memory impairment. Although pharmacology has made considerable progress in these fields, definitive cures have not yet been found for these diseases, and a lot of attention is given to the possibility of finding support in substances of natural origin, at least for their prevention.

Food polyphenols are secondary metabolites of plants, which produce them to defend themselves against bacterial, fungal and viral infections, as well as to resist oxidative stress, heat from the sun and ultraviolet radiation. They occur naturally in many vegetables, herbs, fruits, different types of grains and some beverages derived from them [1,2,3]. Their phenolic structure can vary, and they are often conjugated with sugars. These natural substances, also used as food supplements, have many beneficial effects on health due to their antioxidant properties. They also modulate inflammatory and antimicrobial processes [4,5], which extend to the prevention of pathologies related to metabolic syndrome, cardiovascular diseases, tumors [6,7,8,9] and, as will be seen in this review, some diseases of the nervous system as well.

Since oxidative metabolic and structural anomalies and inappropriate inflammatory reactions play critical roles in the pathophysiology of several brain disorders, including neurodegenerative diseases and ischemic stroke, it is plausible that the antioxidant and mildly anti-inflammatory capabilities of polyphenols could counteract or slow down the development of a wide range of pathologies. This topic is vast and constantly updated, above all because flavonoids have multiple pleiotropic properties (they act on different cellular targets and biochemical mechanisms). Furthermore, the different polyphenols have properties that characterize one molecule more than another, but many of their actions on the organism overlap.

This text therefore represents a review of knowledge in the latter field of investigation, focusing on the topic of the neuropharmacology of polyphenols, with particular reference to two of the most studied flavonoids in the literature on the main degenerative diseases of the nervous system, quercetin and hesperidin. The usefulness of polyphenols has been investigated in population studies associating nutrition habits and disease. However, the large majority of the experimental studies carried out so far are typically of a pre-clinical type, on animal models (mouse, rat, laboratory fish) or on in vitro cells, which establish the pharmacological plausibility of the use of these substances but do not prove it clinically. On the other hand, it is true that demonstrating clinical efficacy in long-lasting chronic degenerative diseases is extremely difficult due to the presence of various comorbidities and a large number of confounding factors. It is equally true that, as pharmacology has progressed, laboratory and pre-clinical studies have always preceded applications in the human field.

The investigation begins with a quantitative evaluation of the literature on polyphenols and neuroprotection; then, the fundamental mechanisms by which flavonoids are thought to protect nerve cells from oxidative stress, toxicity and chronic inflammatory processes are described. Subsequently, the studies carried out so far on the main neurologic and neurodegenerative diseases are described, namely:Anxiety, stress and depression;Neurotoxicity;Alzheimer’s disease;Parkinson’s disease.

Each chapter briefly presents the biological basis of the disorder considered and the progress of human clinical studies and then analyzes the scientific literature, grouping the studies according to the main experimental methods and models while trying to maintain a chronological criterion. The scientific literature is reviewed, dealing with studies on hesperidin and quercetin, in that order, without neglecting information on other polyphenols that have been studied together with them, because some studies refer to multiple components.

## 2. Materials and Methods

A comprehensive literature search was performed to retrieve published articles on the neurological effects of polyphenols and various flavonoids up to 20 December 2022. The original English-language articles were collected from electronic databases (e.g., Scopus, Research Gate, Google Scholar, PubMed and Medline), as well as by cross-referencing the bibliography of the most recent reviews in the field of interest (published in the last decade) [4,5,10,11,12,13,14,15,16,17,18,19,20,21,22,23]. In particular, keywords relevant to the topic of interest were used for this bibliographic search. The words “Polyphenols”, “Flavonoids”, “Hesperidin” or “Hesperetin”, “Quercetin”, “Kaempferol”, “Apigenin”, “Naringenin”, “Taxifolin” and “Catechin” present in the titles of the papers were combined with the following keywords in the abstracts: “Neurodegenerative”, “Neuroinflammation”, “Depression”, “Anxiety”, “Cognitive impairment”, “Alzheimer”, “Parkinson”, “Neuroprotective”, “Antidepressant” and “Anxiolytic”.

An initial quantitative evaluation of the scientific literature in the fields covered by this review was carried out by searching for keywords in the PubMed database, using *EndNote 20*^tm^ (Clarivate, London, UK) software, which allows you to examine the words of interest in the titles and abstracts of the various articles. The results of this search are shown in Table 1.

This evaluation allowed us to estimate a few hundred indexed publications that report the effects of flavonoids in neurology. The pathology topic that is most frequently mentioned in the summaries of the publications on “Flavonoids” (Table 1, second column) is “Alzheimer”, followed by “Neurodegenerative”, “Parkinson”, “Neuroinflammation” and “Depression”. It should be noted that the keyword “Flavonoids” gave similar results to “Polyphenols” (except for “Neurodegenerative”), suggesting that, of all the polyphenols, flavonoids are the most studied molecules in this field.

The flavonoid with the most mentions in these topics is quercetin, followed by hesperidin (or its metabolite hesperetin), naringenin and apigenin, with similar scores, and, finally, kaempferol and taxifolin. Of the therapeutic effects, the word “Neuroprotective” is particularly evident, with quercetin being the most cited substance (312 publications), followed by hesperidin or hesperetin (97), naringenin (60) and apigenin, kaempferol and taxifolina. In the literature, there are also dozens of papers dealing with the antidepressant or anxiolytic effects of flavonoids.

As the keyword, this literature search used the flavonoid of interest in the title and the disease in the summary. This allowed us to focus on single substances and to demonstrate that quercetin and hesperidin are those most cited, also indicating that the choice to focus the study on these two is correct. On the other hand, the research method used does not exclude the same substance being mentioned in the same paper as being of interest for different pathologies (for example, both AD and PD) and having multiple therapeutic properties (for example, neuroprotective and anxiolytic).

We examined the available literature, starting from reading all the abstracts and extracting the complete papers with the criterion of favoring the most recent in the case of similar subjects, while giving equal importance to studies in favor of or critical of the action of the substances studied. Furthermore, we also considered previously published articles in which the basic concepts related to the main topics, such as oxidative stress and the various mechanisms of pathology (e.g., Neurotoxicity, Neuroinflammation, Excitotoxicity, Ischemia-reperfusion), were reported.

The literature on the neuroprotective effects of quercetin and hesperidin is very recent and has grown rapidly over the last 15 years (see Figure 1).

The number of papers on hesperidin and neuroprotection has grown the fastest in the last 10 years and accounts for about half of that for quercetin over the last two years. The neuroprotective effect of hesperidin and hesperetin was first reported by Cho et al. [24]. In cell-free chemical analyses, the authors noted that hesperetin and hesperidin had the ability to scavenge free radicals and that, on primary cultured cortical cells, hesperetin protected against peroxide-induced damage from excess glutamate (excitotoxicity) and from amyloid-beta (Aβ) oligomers. These results demonstrated the potent antioxidant and neuroprotective effects of hesperetin for the first time, suggesting its potential use against various types of adverse factors associated with many neurodegenerative diseases.

As can be seen, the number of articles is not small, but neither is it excessive, and, in the subsequent work, this made it possible to read the results of all the publications and choose to deepen the knowledge of the subject in the publications whose arguments seemed more in line with the research objective. It should also be noted that, while quercetin and hesperidin are the most cited flavonoids as potential remedies against neurological and neurodegenerative diseases, the literature is equally rich in studies on the beneficial effects of other polyphenols of classes other than flavonoids, primarily resveratrol and curcumin, which, however, are not covered by this review.

## 3. Oxidative Stress and Neurotoxicity

Oxidative stress is an important mechanism of cell pathology implicated in many diseases, including those of the cardiovascular and nervous systems. The pathogenesis of several neurodegenerative diseases, including AD, PD and Huntington’s diseases, is strongly associated with oxidative stress [25,26,27,28,29,30,31]. This section summarizes the essential notions of this vast chapter of neuropathology, which is of use for understanding the main targets of the action of flavonoids.

Oxygen is a fundamental substance for maintaining the normal functions of neurons, but its delicate metabolism can be a source of more or less toxic molecular derivatives, and it can affect biological molecules such as proteins, nucleic acids and membrane lipids (Figure 2). These molecular derivatives are considered here under the collective term of reactive oxygen species (ROS) and include O_2_^−^ (superoxide anion), H_2_O_2_ (hydrogen peroxide), ^•^OH (hydroxyl radical), L-O_2_H (lipid peroxide), NO (nitric oxide) and NO_3_^−^ (peroxynitrite).

Under physiological conditions, each cell produces a certain number of ROS through various processes such as redox enzymatic reactions, the oxidative phosphorylation of mitochondria, the metabolism of xenobiotics and the activity of cells of the innate immune defense system. Therefore, many disposal processes protect the delicate cellular balances, and under normal conditions, the cells do not suffer serious damage, other than perhaps slight and progressive molecular alterations which have repercussions on the aging processes. When ROS are in excessive quantities, they react with biological molecules and cause mutations in nucleic acids, damage to proteins (particularly sulfur-containing amino acids) and membrane alterations. All of this can lead to neurodegeneration involving a range of events including mitochondrial dysfunction, Ca(2+) overload, energy deficit and excitotoxicity [17].

Free radicals target almost all the molecules of the organism that undergo phenomena of distortion, resulting in the reduction in or loss of function. DNA (mutations), proteins (enzymatic inactivation, aggregation, degradation) and lipids (lipoperoxidation, membrane rupture, the formation of inflammatory mediators) are particularly sensitive. Lipoperoxidation (Figure 2, right side reactions) is a particularly harmful alteration in the central nervous system because this tissue has an abundance of lipid components in the membranes of nerve cells and, especially, of fibers. During lipid peroxidation, ROS react directly with membrane polyunsaturated fatty acids (PUFAs) to produce toxic aldehydes such as 4-hydroxynonenal (4-HNE) and malondialdehyde (MDA). Iron accelerates this process.

Paradoxically, oxidative stress can also be triggered by oxygen deficiency or by the alternation of ischemia/anoxia and reperfusion. In fact, hypoxia and hypoglycaemia lead to an alteration of the metabolism of the nerve cell, to the extent of depleting (exhausting) the energy reserves (basically, the ATP) and causing the malfunctioning of the mitochondria, with the disorder of the cytochromes and the partial reduction of oxygen. Furthermore, from the metabolism of purines, a considerable amount of hypoxanthine is formed, which, in turn, is oxidized by xanthine oxidase, with the formation of ROS. Oxidative stress then leads to DNA damage, with the need for the repair and consumption of ATP. The main problem lies in the fact that both oxidative stress and ATP depletion have profound consequences for the cell metabolism (ion pump defects, membrane defects, increase in calcium ions, acidosis), which, in turn, trigger the activation of destructive enzymes such as phospholipases and proteases, which ultimately leads to cell death.

### 3.1. Vicious Cycles

In the long run, these alterations can lead to biochemical cell damage and, eventually, cell death, which, in turn, activates new inflammatory processes in the affected tissues and in the nervous system, especially for the involvement of cells of the monocyte–macrophage series and the microglia, which produce ROS. A vicious pathogenic cycle is therefore created between oxidative stress, cytotoxicity and inflammation. In fact, inflammation produces an increase in oxidative stress, and the consequent cellular damage becomes a way to trigger inflammation.

In neurology, vicious circles like these can run a rapid, acute course (such as in stroke or trauma) or a chronic course (such as in AD and PD [28,29]), as well as produce functional neurological disorders. A growing number of studies have shown that depression and anxiety disorders in adulthood could be caused by inflammation at an early age [32]. Elevated levels of neuroinflammatory cytokines such as interleukin (IL)-1β, tumor necrosis factor-α (TNF-alpha) and IL-6 have been identified in patients with depression [33].

One particularly important mechanism connected with infections and inflammation is the production of H_2_O_2_ and O_2_^−^ by the cells that carry out phagocytosis (the engulfing and killing of microorganisms). In fact, these cells (neutrophil granulocytes, eosinophils, monocytes, macrophages, microglial cells) are equipped with a special enzymatic apparatus (NADPH oxidase) capable of generating superoxide and therefore use ROS to kill bacteria and neutralize viruses [34]. However, during this activity, which is in itself defensive and “beneficial”, it is possible that a large number of ROS, accompanied by proteolytic enzymes, are also released outside the cells, causing damage to the nearby cells and tissues. Activated microglia are an important type of innate immune cell in the brain that secrete inflammatory cytokines into the extracellular environment, where they exert neurotoxicity on surrounding neurons by releasing nitric oxide [35,36], and are involved in the pathogenesis of many brain disorders, including AD [37].

Another aspect of notable importance is the presence of “vicious cycles” between a lack of energy (the depletion of ATP), defects of the ion pumps and the activation of the cell, which release neuromediators, such as glutamate, that activate the “excitotoxicity” mechanism. This is particularly dangerous for the nerve cell [38,39,40] and is slowed down by hesperidin [41] and other flavonoids [42,43,44].

### 3.2. Cell Death

Cell death is an extremely complicated pathological process that occurs with various mechanisms that are summarized here in order to introduce the mode of action of the natural compounds covered by this review. Basically, cells die in two ways: by necrosis or by apoptosis. The most “physiological” pathway is apoptosis, which results in cell death without causing damage to the surrounding tissues or significant inflammatory processes, which occurs in the case of necrosis. The characteristics of an apoptotic cell include contraction, membranous blebbing, chromatin condensation and the formation of multiple DNA fragments, ending with the engulfment by macrophages, thereby avoiding an inflammatory response in surrounding tissues. However, if apoptosis occurs at an excessive rate, significant tissue damage can occur, leading to atrophy and functional deficits, phenomena that are often found in neurodegenerative diseases and in the elderly.

In mammalian cells, apoptosis is triggered by the activation of two main pathways: the “intrinsic” pathway, which involves the disruption of the mitochondrial membrane, and the “extrinsic” pathway, which starts from the activation of a pro-apoptotic receptor on the cell surface. The intrinsic pathway is activated by the lack of growth factors or in response to many different damaging influences such as DNA damage, oxidative stress or hypoxia, as well as by many toxic chemical agents such as chemotherapeutics. These agents lead to the release of cytochrome c from the mitochondria, which in turn activates APAF-1 and thus caspase-9. This step can be inhibited by anti-apoptotic members of the Bcl-2 family of apoptosis regulators. Bcl-2 is the prototype of a family of genes and corresponding proteins that govern the permeability of the outer mitochondrial membrane and can be both pro-apoptotic (Bax, BAD, Bak, Bok and others) and anti-apoptotic (Bcl-2, Bcl-xL and Bcl-w, the main ones). Cytochrome c released for mitochondrial damage interacts with APAF-1 caspase-9 and promotes the activation of caspase-3, which in turn acts as the final effector caspase, cutting many cellular proteins including structural proteins, nuclear proteins, cytoskeleton proteins and signaling molecules. Caspases are normally suppressed by inhibitors of apoptosis proteins (IAPs). When a cell receives an apoptotic stimulus, IAPs are silenced by SMAC, a mitochondrial protein, which is released into the cytosol. SMAC binds IAPs, and by binding, it “inhibits the inhibitor” that previously prevented the apoptotic cascade being initiated.

The extrinsic pathway is started by death receptors and involves the activation of caspase-8, which directly activates caspase-3, causing apoptosis. TNF-alpha is mainly produced by macrophages and is the main extrinsic mediator of apoptosis. The Fas receptor is another related receptor. The ligand–receptor interaction results in the formation of a signaling complex that induces cell death, which contains FADD and then caspases. Caspase 8 then activates the cytosolic protein BID by a process of proteolysis. Truncated BID translocates to the mitochondria, facilitates cytochrome c release and activates the intrinsic pathway. The death receptor is regulated by FLIP, which blocks the activation of caspase 8. This process is severely controlled by a number of signaling mechanisms which can also be altered in cancer and which, as will be seen, can be targeted by specific pharmacological agents and nutraceuticals [9,45].

Iron, in its free form, can catalyze ROS reactions (Figure 2), leading to the formation of hydroxyl radicals and lipid peroxides, to glutathione (GSH) depletion, and to the malfunction of antioxidant enzymes and cell death, called “ferroptosis”, which is distinct from apoptosis and necrosis [46]. Morphological features during ferroptosis include smaller mitochondria, an increased membrane density and plasma membrane blebbing. Other cell damage and death mechanisms, particularly those involved in AD and PD, are indicated in the respective chapters.

## 4. Defense and Detoxification Systems

The pathological phenomena associated with ROS are so important and dangerous that cells have evolved to develop defense and detoxification systems, made up of enzymes and natural substances (vitamins A, C and E and uric acid). Furthermore, there are many enzymes involved in protecting against oxidative stress, including superoxide dismutase, catalase and GSH peroxidase. Under physiological conditions, neurons remove oxidation products using different mechanisms. For example, GSH is a potent antioxidant that balances the level of intracellular oxidants by binding to oxidation products and removing them from neurons.

These defense systems are “inducible”, i.e., their production can increase when the transcription and expression of the related genes are activated at the cell nucleus level. It is therefore a functional adaptation capacity for survival. The transcription factor Nuclear factor erythroid 2-related factor 2 (Nrf2) is of primary importance because it regulates gene expression through a promoter sequence known as the antioxidant response element (ARE) [47]. Nrf2/ARE initiates the messenger RNA transcription of a number of target genes, such as those for coding catalase, superoxide dismutase, heme oxygenase (HO-1), GSH peroxidase, thioredoxin and others (Figure 3).

In the nervous system, the HO-1 system has been reported to be highly active, and its modulation appears to play a crucial role in the pathogenesis of neurodegenerative disorders. Many studies clearly demonstrate that the activation of Nrf2 target genes—in particular, HO-1—in astrocytes and neurons is strongly protective against inflammation, oxidative damage and cell death [48]. This biochemical mechanism is activated by various natural substances including polyphenols such as hesperidin and quercetin.

A review [49] demonstrates how pathological events resulting in the overproduction of ROS and the inhibition of the Nrf2/ARE system damage essential cellular components and cause the loss of the structural and functional integrity of neurons. In addition to the Nrf2/ARE antioxidant system, neuronal survival is mediated by neurotrophin signaling via the tropomyosin-related kinase B (TrkB) receptor. TrkB serves as a receptor for the brain-derived neurotrophic factor (BDNF), which is involved in the transcription, translation and trafficking of proteins at various stages of neuronal development and synaptic plasticity, which, in turn, is associated with learning and memory development. The authors propose that antioxidant cellular defense systems, including phytochemicals, be considered drug targets for the treatment of neurodegenerative diseases.

Clearly, medical researchers wondered if it was possible to make cells more resistant to diseases involving ROS as a pathogenic mechanism, so they started looking for drugs or supplements that serve to dispose of excess free radicals [50] or to “boost” the Nrf2/ARE pathway [51,52,53] and therefore to potentially prevent or treat degenerative, viral, immunological and other diseases [28,29].

There are various plants used medicinally with potential benefits in neurodegenerative diseases [13,18,53,54,55,56,57]. Nutraceuticals such as *Hypericum perforatum* (containing hypericin and hyperforin), polyphenols such as hesperidin, quercetin and resveratrol, carnosine and omega-3s provide a concentrated form of bioactive agents that may improve cognitive function and mood alone or in combination with current approved drugs [15,58]. Furthermore, polyphenols have been implicated in neuronal survival by acting on a variety of cell signaling cascades and in synaptic plasticity.

### 4.1. Flavonoids

The class of polyphenols most represented in the Mediterranean diet is that of flavonoids, vegetable pigments which have a basic structure consisting of a skeleton composed of 15 carbon atoms (C-15) arranged in three rings: two benzyl rings—called A and B—and a heterocyclic ring—designated as C (see Figure 4). The word flavonoid comes from the Latin word meaning yellow, as many (but not all) have a yellow color.

Structure–activity relationship studies show that the antioxidant and free radical scavenging properties of flavonoids are due to the ketone group, the double bond between the 2 and 3 carbon atoms, the 3’, 4’-catechol and hydroxyl at the 3-position in the flavonoid skeleton (the last two are present in quercetin but not in hesperidin) [59]. The C2-C3 double bond extends π-conjugation to the carbonyl group in the C ring, so the radical scavenging capacity of unsaturated flavonoids is greater than that of saturated structures, such as flavanones [60]. The anti-radical capacity of flavonols in aqueous solvents is mainly exerted by the mechanism of sequential proton-loss electron transfer, associated with the C3 hydroxyl group, or by electron-proton transfer in the catechol component. Thus, the type of B-ring substitution is also considered to be determinant in the anti-free radical potency of flavonoids [60].

Many of the biological effects of flavonoids appear to be related to their ability to modulate receptors, enzymes and cell signaling cascades rather than a direct antioxidant effect. In fact, the maximum concentrations of flavonoids that can be reached in the blood with very high intakes (~2 µmol/L) are much lower than the concentrations of other antioxidants, such as ascorbic acid (~50 µmol/L), uric acid (200–400 µmol/L) and GSH (700–1500 µmol/L).

The functional interaction between flavonoids and enzymes or receptors occurs through hydrogen bonding and hydrophobic interactions with key amino acids of target proteins. For example, an inhibition of the xanthine oxidase enzyme activity by quercetin is exerted thanks to the C4 and C5 hydroxyl groups [61], and the anti-inflammatory activity depends not only on the number of free hydroxyl groups but also on the methyl group [62].

Of the wide range of flavonoids known for their medicinal (or, rather, nutraceutical) properties, the properties of hesperidin and quercetin are explored here, as these are the most investigated molecules for their neuroprotective effects. Although most of the studies are of a pre-clinical nature, taken together, they suggest that the intake of flavonoids (with the normal diet or with supplements, if necessary) could be a potential intervention for the prevention and/or attenuation of the deterioration of the cognitive decline that accompanies various brain disorders.

### 4.2. Hesperidin and Hesperetin

Hesperidin (3,5,7-trihydroxyflavanone 7-rhamnoglucoside, molecular mass 610.6 g/Mol) is a molecule in which citrus fruits are particularly rich. It is a glycosylated derivative of hesperetin, a flavanone found in fruits such as oranges, tangerines and lemons (see Figure 4). In fresh orange juice, the hesperidin content is about 30 mg per 100 mL [63], but it is found in greater quantities in the white part of the peel (albedo) [64,65]. Traces of hesperidin are also found in propolis [66], in grapes [67] and in other vegetables such as (dandelion) Taraxacum officinale [68].

Hesperidin was isolated for the first time in 1828 by the French chemist Lebreton. The name comes from the word “hesperidium”, the Latin name of the fruit produced by citrus trees. Carl Linnaeus (1707–1778) gave the name *Hesperideæ* to an order containing the genus *Citrus*, an allusion to the golden apples of the garden of the Hesperides.

Hesperetin is the precursor in the synthesis of hesperidin and is also its metabolite, produced in the intestine by transformation by the bacterial flora [69]. Chemically, hesperetin is a trihydroxyflavone with three hydroxyl groups located at positions 3′, 5 and 7, with an additional methoxy substitute also present at position 4. The chemical formula is: C_16_H_14_O_6_; molecular mass 302.3 g/mol.

Flavanones lack a double bond between C2 and C3, and this makes them chiral at the C2 position. Chirality implies that the B ring is not planar, as in flavonols, and is bent in relation to the A–C rings. This difference in molecular orientation is relevant because it can influence the way in which different flavonoids interact with their biological targets and, thus, their bioactive properties.

Hesperidin has long been known to have important neuropharmacological effects, including antidepressant, neuroprotective and memory effects [10]. Hesperetin has been widely reported to exert neuroprotective effects in experimental models of neurodegenerative diseases [5,70], which will be described here. Due to its lipophilic nature, hesperidin can easily cross the blood–brain barrier and provide neuroprotection [71].

Hesperidin has limited bioavailability due to its low water solubility, but in the intestine, it is metabolized to hesperetin, which is more easily absorbed [72]. Unlike hesperetin, which could be absorbed directly in the small intestine, hesperidin, like rutinoside, must pass into the colon and be fermented by the intestinal microflora into an alternative form which is more easily absorbed. Due to this metabolism, hesperidin, which is slowly transformed and absorbed, has a favorable blood half-life (6 h) [72]. In a study of healthy volunteers, orange juice was administered in one dose (8 mL/kg body weight), and blood and urine samples were collected between 0 and 24 h after administration [73]. The peak plasma concentration of hesperetin was 2.2 ± 1.6 µMol/L, with considerable variations in different subjects. The elimination half-life ranged from 1.3 to 2.2 h, indicating short-term kinetics. In another trial [74], after an overnight fast, five healthy volunteers drank 0.5 or 1 L of commercial orange juice, containing 444 mg/L of hesperidin, together with a breakfast without polyphenols. The flavanone metabolites appeared in plasma 3 h after the ingestion of the juice, reached a peak between 5 and 7 h and then returned to the baseline after 24 h. The peak plasma concentration of hesperetin was 0.46 ± 0.07 µMol/L and 1.28 ± 0.13 µMol/L, respectively, after 0.5 and 1 L intakes. The authors concluded that, in a case of a moderate or high consumption of orange juice, flavanones represent an important part of the total polyphenol pool in plasma.

Based on the current evidence, it has been suggested that some of the inconsistent effects of hesperidin in human studies are in part due to the interindividual variability in its bioavailability, which, in turn, is highly dependent on α-rhamnosidase activity and the composition of the intestinal microbiota [75]. Indeed, hesperidin and naringin are metabolized by intestinal bacteria, mainly in the proximal colon, with the formation of their aglycons, hesperetin and naringenin and various other small phenols [76]. Thus, citrus flavanones and their metabolites are able to influence the composition and activity of the microbiota and exert beneficial effects on gastrointestinal function and health. Other bioavailability studies have calculated that, if colon-derived phenolic catabolites are added to the glucuronide and sulfate metabolites, orange juice-derived polyphenols are much more abundant and available than previously thought [77,78,79].

These substances have a wide margin of safety both in nutrition and in supplementation. For example, in an extensive review on the benefits of a diet rich in flavonoids or their supplementation [80], the authors include a supplementation, for periods of at least 4 weeks, of a mixture of hesperidin and naringenin at a dose of 400 mg per day and 500 mg of quercetin. In the form of supplements or nutritional complements, hesperidin is considered harmless, with limited negative effects, due to its non-accumulative nature [16,64]. In animal studies, hesperidin showed a good safety profile [81], with a median lethal dose (LD50) of 4837.5 mg/kg, and in chronic administration, up to 500 mg/kg of flavanone did not induce any abnormalities in body weight, clinical signs and symptoms or changes in blood parameters. The high safety of hesperidin after oral intake was declared by the FASEB (Federation of American Societies for Experimental Biology) at the request of the FDA [82] and was confirmed by animal toxicity studies [81,83] and clinicians [84]. Kumar et al. [85] calculated the toxicity of hesperidin and other flavonoids with respect to rodents, and the LD50 was 12 g/kg, so it is extremely safe. Naringin has an LD of 2.3 g/kg and a reference artificial polyphenol used as a serine protease inhibitor drug; Camostat (with anti-trypsin and anti-plasmin activity) has an LD of 3 g/kg.

### 4.3. Quercetin

Quercetin is a flavonol (3,3′,4′,5,7-Pentahydroxyflavone, molecular mass 302.236 g/Mol) widely present in the vegetable kingdom [86]; an average daily consumption of 25–50 mg is calculated [87], up to about 250 mg per day in “large consumers” of fruit and vegetables [88]. The richest food source is represented by capers, but it is also abundant in onions, kale, broccoli, leeks, berries (e.g., blueberries), grapes and chocolate. In these sources, their content varies from 10 to 100 mg/100 g of fresh weight.

Although the bioavailability of quercetin can vary according to the type of food and according to the intestinal metabolism, by means of dietary supplementation, it is possible to increase the plasma concentration of quercetin [89,90]. For example, plasma concentrations reached 0.43 μmol/L (0.13 μg/mL) after 1 week of supplementation with 150 mg/d of pure quercetin [91], reached 0.63 μmol/L (0.19 μg/mL) after 1 week of supplementation with 80 mg/d of quercetin equivalents from onions [92] and reached a maximum of 1.5 μmol/L (0.45 μg/mL) after 28 days supplementation with high doses of quercetin (>1 g/d) [93].

The free radical scavenging activity of quercetin has been well documented, observing that quercetin exhibits protective effects against oxidative stress-mediated neuronal damage by modulating the expression of Nrf2-dependent antioxidant reactive elements and attenuating neuroinflammation. This is accomplished by suppressing NF-kB signal transduction [11,94,95], counteracting cell lysis and cascading inflammatory pathways [96,97].

Many clinical studies have confirmed the safety of quercetin as a supplement and supported its addition to the diet [98]. This molecule has been added as a supplemental ingredient to the Food and Drug Administration’s Generally Recognized as Safe (GRAS) list [99]. The study by Harwood et al. [100] examined the genotoxic and mutagenic effects of quercetin in different animal and human models. The results confirm the safety of quercetin and the absence of in vivo toxicity and carcinogenic effects.

In published human intervention studies, adverse effects following a supplemental quercetin intake were rarely reported, and those effects were mild in nature [88]. No clinical or epidemiological studies reported compound-related adverse effects after long-term oral quercetin administration (3–1000 mg/day for up to 12 weeks) [100,101]. These results from human clinical trials, along with a long history of safe ingestion, support the idea that quercetin supplementation causes no adverse health effects. In an intervention study [93], healthy men and women took four capsules of a quercetin-containing supplement (1.0 g of quercetin/day) for 28 consecutive days. There were also no alterations in platelet aggregation, platelet thromboxane B2 production, blood pressure or heart rate at rest or the serum levels of total cholesterol, LDL or HDL or triglycerides.

Problems in the use of flavonoids in high doses for a long time can arise from the possible interaction with the main drug transporters (and metabolizing enzymes), leading to alterations in the pharmacokinetics of substrate drugs and, therefore, to their efficacy and toxicity [102]. For example, the bioavailability of the anticancer drug, paclitaxel, was significantly increased in rats with the coadministration of a number of flavonoids, including quercetin, naringin and genistein. However, since paclitaxel is also metabolized by CYP3A and many of the flavonoids, such as quercetin, are reported to inhibit various CYPs, it is likely that the inhibition of both P-gp transport and CYP metabolism may contribute to the improved bioavailability of the paclitaxel. The possible interaction with oral anticoagulants metabolized in the liver and psychotropic drugs should be evaluated with particular care.

## 5. Anxiety, Stress and Depression

Depression is a debilitating mental disorder affecting a substantial number of people globally, hampering all aspects of their lives and resulting in a large number of suicides each year. It is estimated that this challenging condition affects 15–20% of the world’s population [103]. Its origin is multifactorial and intertwines genetic bases, unresolved life stresses, wrong lifestyles and even infectious phenomena and intestinal disorders.

It is impossible to identify one specific factor that causes depression in all patients. Hence, the best approach may be to look for the cause or causes for each individual patient and then apply personalized treatment, not only to relieve depression but also to correct the dysfunction in the body that triggers the depressive symptoms [104].

Depressive syndromes have profound and complex causes, which cannot be reduced to single biological or biochemical mechanisms, but the insufficient activity of monoaminergic neurotransmitters is certainly implicated [105,106,107], as are neuroendocrine axis imbalances (CRH-ACTH-corticosteroids) closely connected to the GABAergic system [108,109,110,111,112] and the consequent different functionality of specific brain areas that regulate sleep, appetite, sexual desire and mood. Chronic psychological or biological stress that the person has not adapted to can lead to an increase in CRH and hyperactivity of the hypothalamus-pituitary-adrenal axis, which does not respond to the feedback that should implement hypercortisolism, probably due to the resistance of cortisol receptors. Furthermore, phenomena of inflammation in the brain are implicated in depressive syndromes [113,114,115,116] and even alterations of the microbiota [112,117], which are modulated by polyphenols [19,118,119,120] (see Figure 5).

The continuous stimulation of the hypothalamic receptors by cytokines, in systemic and chronic inflammatory diseases, can lead to the suppression of these receptors, with a consequent reduction in the response of the adrenergic neuroendocrine axes and, above all, the hypothalamic-pituitary-adrenal axis. A reduction in controls of this type obviously causes the exacerbation of the inflammation itself, and chronic inflammation can facilitate nervous depression. These dynamics have also been analyzed in detail in previous investigations by the author [121,122,123].

Of the most important hypotheses, depression would be associated with a deficit of monoaminergic neurotransmitters such as serotonin (or 5-HT), noradrenaline (or NA) and dopamine (or DA). Therefore, antidepressant therapy must somehow address the shortage of these neurotransmitters. Neurotransmitters are synthesized within the presynaptic nerve ending, stored in vesicles and finally released in the synaptic wall (the space between the presynaptic and postsynaptic nerve endings) in response to certain stimuli. In this way, the transmission of nerve impulses from one neuron to another is made possible. After performing their function, the monoamines are taken up by specific transporters and brought back into the presynaptic nerve ending. At this point, the monoamine oxidases (MAO) intervene, which are the enzymes responsible for the metabolism and degradation of monoamines.

The hypothesis of monoamine depression (dopamine, serotonin and norepinephrine) has been the most widely accepted, and, in fact, antidepressants promote an increase in these molecules in neurons and synapses. However, this theory is not sufficient to explain all mechanisms of depression. Antidepressants that act either by blocking the reuptake of monoamines or by inhibiting their degradation in the synaptic cleft only have an antidepressant effect after a few weeks of treatment. Furthermore, many patients are refractory to available therapeutic drugs, which work primarily by increasing the levels of the monoamine neurotransmitters serotonin and norepinephrine in the synaptic cleft. That chronic derangements and organic pathologies are implicated is also suggested by the fact that neuronal atrophy has been observed in depressed patients and in animal behavior models for depression; it has mainly been detected in the limbic structures that regulate mood and cognition, such as the hippocampus [124].

Both basic and clinical evidence indicate that depression is associated with several structural and neurochemical changes in which levels of neurotrophins, especially brain-derived neurotrophic factor (BDNF), are altered [124]. Antidepressants, as well as other therapeutic strategies, appear to restore these levels. Furthermore, chronic antidepressant treatment improves adult hippocampal neurogenesis, supporting the idea that this event underlies the effects of antidepressants.

Furthermore, a subset of patients who are unresponsive to antidepressant medications show increased systemic and central inflammatory responses, which collectively led to the evolution of the inflammatory theory of depression. Indeed, inflammatory mediators generated in the periphery, as well as insults within the brain, can activate brain-resident immune cells, resulting in a neuroinflammatory response that interferes with the multitude of neurobiological mechanisms implicated in the pathogenesis of depression.

Eating habits are closely linked to depression [125], with the Mediterranean diet being more beneficial than the average Western diet [126]. A meta-analysis has shown that [127] the risk of depression is higher in population groups that adopt dietary models characterized by refined cereals, sweets, animal fats, potatoes and high-fat sauces compared to models characterized by a high consumption of vegetables and fruits, whole grains, fish, olive oil and low-fat and antioxidant dairy products and by a low consumption of foods of animal origin. A dietary pattern characterized by a high consumption of red and/or processed meat, refined grains, sweets, high-fat dairy products, butter, potatoes and high-fat sauces and a low consumption of fruits and vegetables is associated with an increased risk of depression.

The antidepressant potential of some important polyphenols such as amentoflavone, apigenin, chlorogenic acid, curcumin, ferulic acid, hesperidin, rutin, quercetin, naringenin, resveratrol, ellagic acid, nobiletin and proanthocyanidins offers an interesting perspective in the complementary treatment of depression [128]. Recent evidence suggests that the diet modifies key biological factors associated with the development of depression. It has been suggested that this may be due to the high flavonoid content commonly found in many plant foods, beverages and dietary supplements [129]. Polyphenols have been shown to exert neuroprotective functions on the brain by influencing a number of neuropathological mechanisms, including neuroinflammation [19,130,131].

The anxiolytic effects of quercetin have also been attributed to the inhibition of GABAergic excitation [132,133,134,135]. A review of the antidepressant effects of quercetin was recently published [21]. Similar to what was seen for hesperidin, quercetin’s action mechanisms involve regulating neurotransmitter levels, promoting the regeneration of hippocampal neurons, ameliorating hypothalamic-pituitary-adrenal (HPA) axis dysfunction and reducing inflammatory states and oxidative stress.

Table 2 summarizes the main experimental studies on this subject, with the related bibliography and doses used, in chronological order, first for hesperidin and then for quercetin. Unless otherwise indicated, dosages refer to oral administration.

Some particularly significant and illustrative studies of the experiments on the effects of hesperidin and quercetin on models of depression, post-traumatic and stress disorders are summarized below, grouped by topic.

### 5.1. Traditional Medicine

The first studies of the neurological effects of hesperidin started from traditional medicine and, particularly, from studies on valerian, used as a natural sedative. Valeriana officinalis is a popular tranquilizer that has been known in traditional medicine for centuries. Its active ingredients were assumed to be terpenoids in the form of valepotriates and/or essential oil components. However, back in 2003, Marder and collaborators [136] demonstrated that the plant also contains three glycosidic flavones: linarin, hesperidin and methyl-apigenin, and they described its sedative and sleep-improving properties in experimental mouse models. The same group reported that these three substances have synergistic sedative and sleep-enhancing actions, which are enhanced by the simultaneous administration of valerenic acid or diazepam [137]. With various bioinformatic analyses of the proteins transcribed from neuronal genes and their ligands, Santos and collaborators [143] analyzed some components of the Valeriana officinalis extract and suggested that hesperidin, its main flavonoid, along with linarin, are able to alleviate the effects of oxidative stress during ATP depletion due to their ability to bind sulfonylurea receptor-1 (SUR1), the regulatory subunit of K-ATP channels, expressed in neurons, astrocytes, oligodendrocytes, endothelial cells and in microglia. The authors attribute the protection of the neuronal function of valerian in conditions of hyperexcitability, oxidative stress, ischemia and/or ATP depletion to these properties of hesperidin and linarin.

An infusion made from the flowers of several species of the Citrus genera is used as a sedative to treat insomnia in traditional Mexican medicine [140]. Byrsonima crassifolia (Malpighiaceae) is also a plant used in traditional medicine for its anxiolytic, anticonvulsant and antidepressant effects. The results of a study on mice [141] show that the methanolic extract of Byrsonima crassifolia, rich in flavonoids such as rutin, quercetin and hesperidin, may be involved in the antidepressant effects. Pharmacokinetic tests in laboratory mice have ruled out that the effects of hesperidin include a direct action on the benzodiazepine binding site and have suggested that this flavonoid could be used to decrease the effective therapeutic doses of the same benzodiazepines [138]. Subsequent studies have described the synergistic properties of hesperidin with other benzodiazepines [139]. The same research group subsequently demonstrated that the systemic administration of hesperidin in sedative doses produced a marked reduction in the phosphorylation status of extracellular signal-regulated kinases (ERK 1/2) [180]. Extracellular signal-regulated kinases (ERKs) are signaling molecules that are involved in the regulation of many functions activated by various stimuli, including growth factors, cytokines, virus infection and ligands for G protein-coupled receptors.

Immature orange fruit (*Citrus aurantium* L.) is widely used in traditional Chinese medicine, in decoctions formulated to treat depressive syndromes, such as the *Chaihu Shugansan* [181]. A review of Chinese authors [21] retrieved clinical trials on depression using Chinese herbal formulas as interventions in which quercetin was the main component. For example, the effects of *Chaihu Shugansan* in improving depression when used as a monotherapy were significantly better than antidepressants such as fluoxetine, paroxetine and duloxetine and were comparable to these antidepressants in improving the cure rate. Therefore, according to the authors, quercetin may work synergistically with other components to improve depression.

Among other things, it is interesting that quercetin is one of the components of *Hypericum*, a plant whose antidepressant power is ascertained and is normally attributed to hypericin.

### 5.2. Post-Traumatic Stress Models

Purified hesperidin and quercetin have been studied in several experiments of stress and anxiety models. Given the vastness of the literature, the most salient and recent results are reported here, with the intention of describing the variety of experimental models used. Post-traumatic stress disorder is a psychiatric disorder characterized by depression and anxiety, which arises due to an imbalance of neurotransmitters in response to excessive stress. There is growing evidence that traumatic brain injury is associated with an increase in depression-related disorders, so some authors have asked the question of whether the dietary intake or the supplementation of natural flavonoids such as hesperidin and quercetin can be used as a therapy for patients with brain injury and depression. The main models are those of the standard behavioral tests in mice and rats.

Various authors have demonstrated, in experimental models in rodents (e.g., forced swimming and suspension on the tail), that hesperidin has a significant antidepressant effect, probably linked to its modulatory potential on 5-HT1A serotonin receptors [144]. The administration of hesperidin and fluoxetine, at sub-effective doses (less than 1 mg/kg, intraperitoneally), produced a synergistic antidepressant effect. The results of Brazilian researchers [145,182] showed that treatment with hesperidin or hesperetin has an antidepressant effect in the rat tail suspension test without changing the locomotor activity in the open field test. This effect of hesperidin is probably mediated by the inhibition of the L-arginine-nitric oxide (NO)-cyclic guanosine monophosphate (cGMP) pathway and by increasing BDNF levels in the hippocampus.

Hesperidin has been proven to be effective in post-traumatic depression models [146,150,157]. Interestingly, hesperidin treatment reduced the levels of IL-1beta, TNF-alpha and MDA (oxidative stress marker) and increased BDNF levels in the hippocampus. Furthermore, oxidative stress-related molecules, including superoxide dismutase, catalase, Nrf2 factor and heme oxygenase-1, were increased by hesperidin treatment. In this context, it is interesting that hesperidin has shown in vivo antidepressant effects in laboratory animals, with mechanisms involving nerve signal transduction pathways [152] and the modulation of 5-HT concentrations [154].

Another classic rodent model of depression is olfactory bulbectomy (OBX), which results in several behavioral and biochemical changes, useful as a screening model for antidepressants. In an experimental rat study, the ablation of the olfactory bulbs caused hyperactivity in the open field arena and increased the immobility time in the forced swim test, which was coupled with increased serum corticosterone levels [165]. After a 2-week surgical recovery period, quercetin treatment significantly decreased the behavioral, biochemical, molecular and histopathological changes induced by OBX. Notably, there was the normalization of oxidative-nitrosative stress markers, inflammatory mediators (TNF-alpha and IL-6) and caspase-3 (an apoptotic factor) in both the cerebral cortex and hippocampus.

### 5.3. Psycho-Social Stress

Imbalances of neuroimmune, neurotrophic and neurochemical homeostasis have important implications in psychopathologies resulting from psychosocial maladjustment. Bhutada et al. [164] investigated the influence of quercetin on CRF (corticotropin-releasing factor) or CRF antagonist (antalarmin)-induced changes in the social interaction time in the social interaction test and in the immobility time in the forced swimming test. The results indicated that both the quercetin and antalarmin doses dependently increased the social interaction time and decreased the immobility time, indicating an anxiolytic-like and antidepressant effect. Another study also demonstrated that anxiety, depression and cognitive impairments induced by psychosocial stress are ameliorated by quercetin and also by modulating plasma corticosterone and adrenocorticotropic hormone levels, as well as corticotropin-releasing factor (CRF) mRNA expression in the hypothalamic region [183]. One of the action mechanisms considered for this effect of quercetin is the inhibition of monoamine oxidase activity, which results in the best preservation of dopamine and serotonin rates [184,185].

A classic model for testing the antidepressant effects of drugs is chronic mild unpredictable stress (CUMS), which is closest to human depression. This model involves subjecting mice to daily repeated mild stress such as changing the light/dark cycle several times or changing the chipboard bottom of the cage numerous times. The group of Guan and collaborators [172,173] demonstrated an antidepressant effect of quercetin in vivo on a CUMS model carried out for 21 consecutive days. This stress caused significant decreases in sucrose preference in behavioral tests in an open field and forced swimming tests, prevented by quercetin in a dose-dependent manner. In the depressed group, the serum iron, copper and calcium levels increased significantly, while the magnesium, zinc, selenium and cobalt levels decreased significantly. The levels of the above-mentioned seven elements were significantly normalized by quercetin, and indices of oxidative stress improved. In the study by Khan and collaborators [43], the administration of quercetin was started 2 weeks after the start of the CUMS protocol and continued for up to 6 weeks. Stress caused a decrease in the levels of antioxidant markers and 5 HT in the brain tissue, changes significantly attenuated by quercetin.

Another model of chronic stress in mice, called the social defeat stress (SDS), consists of subjecting an experimental animal to repeated social subordination by an aggressor male placed in the same cage. Zhang and collaborators [170] studied the effect of a quercetin-enriched diet on SDS-induced depressive behaviors. The quercetin-enriched diet, when introduced before the onset of stress conditioning, attenuated the depressive-like behavior of the mice. The researchers also found that astrocyte activation in the hippocampus decreased during the long-term administration of the quercetin-enriched diet. In the SDS model, decreases in prefrontal cortex serotonin and striatal dopamine elevated norepinephrine and acetylcholinesterase (AChE) levels in the prefrontal cortex and hippocampus, with corresponding decreases in BDNF, which were reversed by quercetin [178]. Adrenal hypertrophy, increased blood glucose and corticosterone release were reduced by quercetin, suggesting that quercetin attenuates the consequences of psychosocial stress by normalizing the hypothalamic–pituitary–adrenal axis, modulating neurotransmitter release, enhancing BDNF and inhibiting neuroinflammation. According to other authors, however, the antidepressant effect of quercetin would occur even in the absence of demonstrable changes to the neuroendocrine axis [167].

Methamphetamine (MA) abuse also causes neurotoxic outcomes, including increased anxiety and depression. Therefore, a model of anxious behavior in mice induced by AD administration was developed [177]. Quercetin exerted antipsychotic activity, alleviated anxiety-like behavior and ameliorated mitochondrial impairment by decreasing the reactive oxygen species levels and mitochondrial membrane potential and increasing ATP production. Furthermore, the study examined the effect of quercetin on astrocyte activation and neuroinflammation, and the results indicated that it significantly attenuates astrocyte activation and reduces the levels of IL-1beta and TNF-alpha, but not IL-6.

### 5.4. Depression and Diabetes

Depression is highly prevalent in diabetics and is associated with poor glucose regulation and an increased risk of diabetic complications, such as neuropathy. The identification and effective treatment of co-occurring depression are increasingly seen as essential components of the clinical care of diabetics. Oxidative stress has been implicated in the pathogenesis of streptozotocin-induced diabetes mellitus (STZ) and its complications in the central nervous system. An initial study on this topic examined the hypothesis that hesperidin could be a factor that limits the complications of diabetes [142]. Diabetes mellitus was induced in rats by a single injection of STZ, and three days later, the administration of hesperidin was started for 4 weeks. The treatment significantly attenuated the altered levels of lipoperoxidation (thiobarbituric acid) and biomarkers of neurotoxicity (brain GSH and AChE activity).

Diabetic neuropathy can also be induced in rats by the administration of a high-fat diet plus injections of STZ [186]. Animals treated in this way developed neuropathy after a few weeks, but if hesperidin (100 mg/kg for 1 month) was added to the diet, the disease was slowed down. Concomitantly, the authors observed an increase in antioxidant enzymes, an upregulation of SIRT1 and an inhibition of NADPH oxidase 4 (NOX4).

A series of studies on the effects of hesperetin against anxiety and depressive disorders caused by streptozotocin-induced diabetes explored the potential mechanisms related to the activation of the Nrf2/ARE pathway, which serves the activation of major antioxidant systems [159,160]. The results show that hesperidin supplementation also caused an increased expression of protein kinase A, CREB (cAMP response element-binding protein, a cellular transcription factor) and BDNF in the amygdala and hippocampus.

The antidepressant activity of quercetin was also evaluated using the forced swimming-induced behavioral hopelessness test in the mouse model of STZ-induced diabetes [161]. Quercetin dose-dependently reduced the period of immobility in diabetic mice, and this effect was comparable to that of fluoxetine and imipramine. Positive effects of quercetin on STZ-induced depression-like behavior in rats have also been reported by others [167].

The exposure of mouse brain endothelial cells (bEnd3) to hyperglycemia in vitro causes an increased production of ROS [187], with an increased expression of NOX4 and nitric oxide synthase (eNOS), which are attenuated by the polyphenols of medicinal plants, including quercetin.

### 5.5. Depression and Neuroinflammation

Depression and neuroinflammation are closely linked, and flavonoids may also have protective effects due to their ability to modulate inflammatory processes taking place in the brain or systemically. One of the first papers to demonstrate this was by Soliman in 1998 [188]. The authors used an astrocyte culture-based model of neuroinflammation in which nitric oxide (NO) production was stimulated by LPS. In this experiment, the following compounds showed a dose-dependent suppressive effect on NO production: quercetin, epigallocatechin gallate, morin, curcumin, apigenin, sesamol, chlorogenic acid, fisetin, taxifolin, ellagic acid and caffeic acid.

Even at an experimental level, there is accumulating evidence of the deleterious effect of LPS on cerebral inflammation and depression, along with the positive effects of flavonoids, both by direct action on central mechanisms and by indirect action on the regulation of intestinal bacterial flora and barrier functions. In particular, we note the importance of the microbiome and the permeability of the intestinal wall, because the bacterial endotoxin can contribute to neuroinflammation and neurodegeneration if it reaches the systemic circulation and the brain [189,190]. Interactions between the bacterial microbiome and innate and adaptive immune cells trigger autoreactivity, chronic inflammation and tissue damage in genetically susceptible individuals [191,192]. This can occur through the modification of substances by the intestinal bacterial flora, which can therefore become self-antigens and erroneously activate immune responses of the wall itself. Furthermore, recent studies highlight that the disruption or increased permeability of the intestinal barrier and the translocation of commensal bacteria or endotoxins to non-intestinal organs can initiate several autoimmune pathways [193].

In mice, depressive-like conduct can be induced by injections of LPS, which causes a state of neuroinflammation and neuronal oxidative stress, accompanied by memory impairment and a reduction in neuroplasticity. The effects of hesperidin on LPS-induced depressive-like behavior in mice were investigated [148]. LPS injection decreased cravings for sucrose and increased serum corticosterone levels, but it also elevated IL-1beta, IL-6 and TNF-alpha in the prefrontal cortex. Furthermore, LPS downregulated the expression of miRNA-132, a noncoding RNA molecule involved in neurodevelopment, synaptic transmission, inflammation and angiogenesis. Pre-treatment with hesperidin prevented these LPS injection-induced abnormalities. In the study by Nandeesh et al., LPS treatment caused depressive-like behavior, with a worsening of tests such as maze movement, forced swimming and open field movement, accompanied by a decrease in tissue GSH and significantly higher levels of lipid peroxides and IL-6 compared to the control level [151]. Pre-treatment for 3 days with the fraction enriched in orange peel polyphenols showed significant improvements in LPS-induced behavioral, anorexic and biochemical parameters. Similar effects were obtained with dexamethasone (1 mg/kg, i.p), suggesting that they are mediated by inflammatory reactions. Molecular docking and dynamic studies against NF-kappaB (involved in the production of inflammatory cytokines) were also performed, demonstrating that hesperidin has favorable docking scores and interacts with as many as five amino acids of this factor. Similar results have also been obtained by others with hesperetin [149], quercetin [169] and quercitrin (glycosylated quercetin with rhamnose) [174]. In the latter experimental investigation, the antidepressant effect appeared 2 h after the administration and lasted for at least 3 days, and quercitrin reduced the levels of inflammation-related factors including IL-10, IL-1beta and TNF-alpha in serum, as well as activations of the PI3K/AKT/NF-kappaB and MEK/ERK pathways in the hippocampus. Expressions of pCREB/BDNF/PSD95/Synapsin1 neuroplasticity signaling molecules were upregulated.

Microglial cells play a key role in neurodegenerative processes. In LPS-stimulated microglial cells, hesperetin strongly inhibited nitric oxide production and inducible nitric oxide synthase expression [153]. In the same cells, hesperetin also significantly reduced the secretion of inflammatory cytokines including IL-1beta and IL-6. Quercetin also has an inhibitory effect on NO production by LPS-stimulated microglial cells, accompanied by the downregulation of extracellular signaling pathways and the NF-kB factor [194]. In addition, quercetin scavenged free radicals and produced inhibitory effects on serine/threonine and tyrosine phosphatase activities. In other words, the action highlighted by these authors is clearly both antioxidant and anti-inflammatory. An experimental investigation on glial and neuronal cells in vitro was published by Canadian researchers in a neuroscience journal [163]. The authors cultured N9 glial cells and treated them with LPS, inducing the expression of IL-1alpha and TNF-alpha. Pre-treatments with quercetin or resveratrol significantly inhibited the cellular response to LPS. Furthermore, they undertook a co-culture of microglia with dopaminergic neurons (PC12 lineage) to examine the influence of resveratrol and quercetin on LPS-activated microglia-evoked neuronal cell death. The treatment of N9 microglial cells with resveratrol or quercetin successfully reduced the apoptotic death of neuronal cells in the co-culture system, as detected by caspase-3 measurement.

To investigate this aspect, the authors cultured primary microglia cell lines (derived from mice) and challenged them with various toxic factors in the absence and presence of quercetin [175]. The flavonoid significantly attenuated LPS-induced inflammatory factor production, cell proliferation, NF-kappaB activation and inducible nitric oxide synthetase (iNOS) function. Quercetin increased the expression of Nrf2 and reduced the levels of the NLRP3 inflammasome. Remember that the inflammasome is a multimeric protein complex that senses harm or harm-associated molecular patterns (DAMPs) and serves as a major platform for caspase-1 activation and the maturation of the proinflammatory cytokine IL-1beta. NLRP3 inflammasome is the most highly expressed member in microglia, and the excessive activation of the NLRP3 inflammasome is involved in the pathogenesis of many brain disorders, including trauma, ischemia and major depressive disorder, as well as PD [195]. The study cited [175] also demonstrates that quercetin treatment alleviates neurodegeneration in mice models of depression, which is achieved by administering LPS and IL-1beta.

### 5.6. The Importance of Microbiota

As we have seen, LPS is widely used in experimental depression and neuroinflammation models, but the fact that the bacterial endotoxins of Gram-negative bacteria are predominantly of gastrointestinal origin and become factors of clinical complications in cases of the increased permeability of the barrier assumes practical relevance in human pathology. Today, we speak of the “brain–gut connection” or “gut–brain axis” [196] to indicate the complex neuroendocrine relationships between the two organs and, therefore, the importance of the health and proper functioning of the intestine in maintaining the health and proper functioning of the brain. The synergism between LPS and inflammatory cytokines is one of the simplest and most ubiquitous mechanisms of neuroinflammation and neurodegeneration [189,197].

The microbiota–gut–brain axis is a dynamic system of tissues and organs including the brain, the endocrine glands, the immune cells, the gut and the gastrointestinal microbiota, which communicate in complex and multidirectional ways to maintain homeostasis. Changes in this environment can lead to a broad spectrum of physiological and behavioral effects including the activation of the hypothalamic–pituitary–adrenal (HPA) axis and the altered activity of neurotransmitter systems and immune function. The passage of LPS from the intestine to the lymph or to the blood, due to an increase in the permeability of the intestinal barrier, can certainly complicate the progress of acute and chronic inflammatory processes, possibly due to other etiopathogenetic mechanisms. This new evidence has led some researchers to argue that alterations in the gut microbiome play a pathological role in various neurological diseases, including anxiety and depression, as well as chronic pain, schizophrenia [198], dementia [190] and autism spectrum disorders [199,200].

The alteration of the normal commensal intestinal microbiota is also referred to by the generic term dysbiosis. Dysbiosis is associated with diarrhea or constipation, which causes a state of intestinal inflammation (even chronic), which, in turn, is fundamental in promoting endotoxemia, systemic inflammation and neuroinflammation [190]. Obstinate constipation is associated with dysbiosis and the passage of endotoxins (lipopolysaccharide) from the intestine to the blood and can be corrected with prebiotics [201]. Systemic inflammation related to the pathogenic gut microbiota (due to barrier dysfunction, the elevation of lipopolysaccharide and pro-inflammatory cytokines) can trigger or worsen neuroinflammation in brain regions including the hippocampus and cerebellum.

Considering the bidirectional communications between the intestine and the brain, a study examined the changes in the intestinal bacterial flora following the intake of flavonoid-rich orange juice [119] in a group of subjects with symptoms of depression. At the same time, the effect of a dietary change on symptoms of depression was evaluated. For 8 weeks, participants consumed flavonoid-enriched orange juice (600 mg per day, mostly hesperidin, followed by apigenin and naringenin) or a drink of a similar color and flavor, but with only 100 mg of flavonoids. Those who consumed a drink rich in flavonoids showed an increase in bacteria from the *Lacnospiraceae*, *Bifidobacteriaceae* and *Akkermansiaceae* families. Depressive symptom scores using a validated scale showed a marked improvement with flavonoids in young adults. Beneficial effects on the neuroendocrine axis and microbiota have also been described for other polyphenols, including quercetin and resveratrol [118,202,203].

Taken together, these results suggest that hesperidin and quercetin possess some pharmacological antidepressant-like properties and could be of interest as complementary therapeutic agents for the treatment of depressive disorders, elicited by a variety of stressors by modulating the gut-brain axis as well.

## 6. Neurotoxicity

There is a great deal of evidence that flavonoids—primarily, hesperidin, hesperetin and quercetin—have protective effects on the nervous system, when it is subjected to exogenous neurotoxic type aggressions, or as a consequence of endogenous toxicity dependent on metabolic disorders (diabetes) or mediators of the inflammation. Some of these studies have already been mentioned in the previous chapter, where the depressive effects of LPS were discussed. On the other hand, even if the origin of the lesion is a toxic insult, the phenomenon cannot be dissociated from the inflammatory reaction of the accessory cells (astrocytes and microglia), which, in turn, can become a factor in pathology for the release of toxic molecules derived from oxygen in a vicious circle.

Table 3 summarizes the main in vitro and in vivo experimental investigations, with the related bibliography and doses used.

The protective effect of hesperidin was tested against ischemia-reperfusion injury in the rat brain area in a model consisting of bilateral common carotid artery occlusion followed by reperfusion [204,227]. Pre-treatment with hesperidin significantly improved neurobehavioral impairments (neurological score, locomotor activity, lateral push resistance), attenuated oxidative damage and restored the activities of antioxidant and mitochondrial complex enzymes in the brain. More recently, other authors have noted positive effects of hesperidin in mice with cerebral ischemia induced by middle cerebral artery occlusion, showing blood–brain barrier protection and a reduction in MMP-3/9 (a type of protease) [213].

Unsurprisingly, the evidence of the neuronal toxicity of ischemia is derived, above all, from studies on experimental animals, but some evidence is beginning to emerge in the human field as well. Indeed, hesperidin has been used in a clinical context in the treatment of strokes [228]. Treatment with recombinant tissue plasminogen activator (rt-PA) is currently the most effective therapeutic option against cerebral ischemic stroke, but the high incidence of symptomatic intracerebral hemorrhage greatly hinders the ideal treatment outcome of rt- PA. The authors sought to evaluate the impacts of hesperidin on symptomatic intracerebral hemorrhage following rt-PA therapies. Patients with ischemic stroke were randomly assigned to two groups, to receive either rt-PA + placebo (cellulose) or rt-PA + hesperidin. The combined treatment of rt-PA with hesperidin produced a significant improvement in outcomes, as revealed by the better transcranial Doppler ultrasound and NIH Stroke Scale scores, as well as an increase in TGF-beta and a decrease in serum MMP-2 levels and MMP-9 (metalloproteinase). The authors suggested the potential clinical application of hesperidin supplementation to improve the outcomes of rt-PA treatment in ischemic stroke patients.

Excitotoxicity is a phenomenon of neuronal damage that results from the exposure to relatively high concentrations of glutamic acid (50–100 µM). The phenomenon is particularly important because glutamate is the main excitatory neurotransmitter in the central nervous system. In rat hippocampal nerve terminals, hesperidin inhibited glutamate release and the elevation of cytosolic free Ca^2+^ concentration evoked by 4-aminopyridine [41]. Furthermore, the intraperitoneal injection of kainic acid elevated extracellular glutamate levels and caused a marked neuronal loss in the hippocampus. These cainic acid-induced changes were attenuated by pre-treatment with hesperidin.

Aluminum is a light and toxic metal that is ubiquitous on Earth and has gained considerable attention due to its neurotoxic effects, in which the production of ROS is also implicated. It has also been linked ecologically and epidemiologically to several neurological disorders, including AD, PD and amyotrophic lateral sclerosis [220,229]. Some adverse effects of vaccinations have also been attributed to the presence of aluminum oxides in adjuvants [230,231], possibly due to a genetic background [232], so much so that some authors have proposed investigating vaccines with adjuvants of different types [233,234,235].

The effect of hesperidin on aluminum chloride (AlCl_3_)-induced neurotoxicity in mice has been studied by some authors [207]. Six weeks of treatment with AlCl_3_ caused nitrosative oxide stress (lipid peroxidation), cognitive impairment, an increase in proinflammatory cytokines (IL-1beta and TNF-alpha) and increased AChE activity and reduced the BDNF content in the hippocampus of animals. However, chronic treatment with hesperidin (as well as silibinin, a flavonoid component of milk thistle) significantly ameliorated the AlCl3-induced cognitive impairment and hippocampal biochemical abnormalities. Sharma and collaborators [220] demonstrated that quercetin administration reduced aluminum-induced neurodegenerative changes in laboratory rats, ROS production, DNA fragmentation and biochemical mechanisms of apoptosis, while it increased mitochondrial superoxide dismutase activity. Further electron microscopic studies revealed that quercetin attenuates aluminum-induced mitochondrial swelling, cristae loss and chromatin condensation. Another study [222] observed that the aluminum chloride treatment of rats resulted in a significant increase in lipid peroxidation, protein carbonyl levels and acetylcholine esterase activity in the brain. This was accompanied by a significant decrease in GSH, GSH reductase and superoxide dismutase activities. The pre-treatment of rats with quercetin or alpha-lipoic acid resulted in a tendency towards the normalization of most parameters, with a synergistic effect of the two molecules in protecting the rat brain from aluminum chloride-induced oxidative stress. Other studies on the protection from aluminum toxicity by flavonoids are reported in the chapter on AD.

Another interesting model is neurotoxicity from hyperhomocysteinemia, a typical biochemical disorder that can be involved in several human pathologies, including neurological ones [236]. Hesperidin showed significant dose-dependent protective activity against vascular dementia in the rat model of l-methionine-induced hyperhomocysteinemia [210]. Furthermore, hesperidin significantly reduced endothelial dysfunction by decreasing nitrite, serum homocysteine and malondialdehyde levels, along with AChE activity, and increasing superoxide dismutase, GSH and catalase levels.

Neuroprotective effects with the antioxidant mechanism of flavonoids have also been observed against cadmium-induced toxic alterations [208,209,219,237,238,239], azidothymidine (AZT) [223], vincristine [224], methotrexate [215], arsenic (an environmental contaminant) [216], fluoride [32,214], emamectin benzoate (an insecticide) [217] and excess tissue iron [225,240], as summarized in Table 3. Hesperidin has also shown efficacy in the case of acute neurological pathologies such as experimental hepatic encephalopathy caused by thioacetamide [241], restoring the redox balance, as demonstrated by the decrease in MDA in both the liver tissue and the brain by increasing sirtuin-1 expression and decreasing NLPR3 and IL-1beta.

Some systematic reviews have collected the studies conducted on hesperetin in both in vivo and in vitro models of neurodegeneration [5,16,242]. The results demonstrated consistent neuroprotective effects of hesperetin against different models of neurodegeneration and illustrated the multiple underlying mechanisms. The neuropharmacological potential of hesperidin includes anticonvulsant, antidepressant, antioxidant, anti-inflammatory, locomotor and memory enhancement activities.

## 7. Alzheimer’s Disease

Alzheimer’s disease (AD) is characterized by the gradual damage of cells in the brain and spinal cord associated with sensory dysfunction, dementia, functional loss (ataxia and global cognitive deficits), brain atrophy or even death. With the lengthening of the average lifespan and the type of life and nutrition in the richest countries, AD is becoming a serious public health problem all over the world, and the search for natural pharmacological remedies is very active. It has been argued that, if the onset of the disease could be delayed by a decade, the number of AD victims would be decidedly reduced [95].

Deficient cholinergic transmission, amyloid plaque deposition and neurofibrillary tangles (NFTs), chronic neuroinflammation resulting in the progressive degeneration and/or death of nerve cells and cognitive impairment are seen in AD (see Figure 6). Two common pathological hallmarks of AD are senile plaques containing aggregates of extracellular Aβ oligomers and NFTs containing aggregates of the abnormally hyperphosphorylated tau protein.

The transition from normal cognition to dementia over decades is aided by acquired and genetic risk factors. Aβ is produced from amyloid precursor protein (APP in Figure 6) by the action of γ secretase and β secretase as monomers, but these can then form soluble oligomers, which eventually form extracellular precipitates such as amyloid plaques. Beta-secretase is an enzyme that functions upstream in the amyloidogenic processing of APP to generate the A-beta protein that rapidly aggregates to form fibrils, the most abundant component of the plaques seen in the brains of AD patients.

Aβ oligomers inhibit the glutamate uptake by astrocytes, thereby potentiating the action of synaptically released glutamate. This, coupled with a loss in GABAergic inhibition, leads some neurons to become hyperexcitable. Meanwhile, Aβ oligomers also induce the hyperphosphorylation of the axonal microtubule-associated protein tau, which leads to the redistribution of tau into dendrites, where it disrupts the trafficking of glutamate receptors and thereby depresses neuronal excitation and function. These synaptic effects and Aβ- and/or tau-induced axonal myelin loss can induce cognitive dysfunction before synapses are lost and neurons die. The early events in AD include a decrease in cerebral blood flow, probably caused by oligomeric Aβ. Falling oxygenation potentiates neurodegeneration by promoting the hyperphosphorylation of the tau protein [243].

In addition, abnormal iron levels have been reported in the hippocampus and cortex of subjects with AD, and there is a relationship between iron accumulation and pathological features [240]. On the other hand, 4-HNE generated by lipid peroxidation, itself catalyzed by iron, reacts directly with Aβ and produces oxidation products, leading to the aggregation of Aβ. Furthermore, iron can bind to Aβ and increase its aggregation. According to these explanations, it can be concluded that there is a vicious cycle between iron accumulation, oxidative stress, Aβ aggregation and tau hyperphosphorylation.

Unfortunately, there is no cure for AD, and many medications aim to relieve symptoms. Therapies targeting late events in AD, including the aggregation of Aβ and hyperphosphorylated tau, have largely failed, possibly because they are administered after significant neuronal damage has occurred. Since the main cognitive deficits are attributed to cholinergic connections, the drugs that are used in this field are based on the inhibition of AChE. Cholinesterases are reversibly inhibited by many substances, including the alkaloids eserine, prostigmine and rivastigmine, used in pharmacology.

Naturally occurring flavonoids have received considerable attention as alternative candidates for AD therapy, taking into account their antiamyloidogenic, antioxidant and anti-inflammatory properties. Accumulating evidence suggests that they can potentially improve memory and cognitive function and prevent neurodegeneration [4,244] or depression [245]. Among the most promising groups of flavonoids are myricetin, rutin, quercetin, fisetin, kaempferol, apigenin and hesperidin, which have been shown, in vitro, to possess antiamyloidogenic and fibril destabilizing activities, and they are able to act as chelators of metals and to suppress oxidative stress [14,18,95,246,247,248,249].

A retrospective study determined the total dietary intake of Finns for the year [250]. The intake of flavonoids in food was estimated and compared with the incidence of diseases considered by different national public health registries. People with a higher hesperetin intake had a lower incidence of cerebrovascular disease and bronchial asthma. Research from 2015 indicates that orange flavanone consumption is associated with cognitive benefits in adults with mild cognitive impairment and neurodegenerative diseases [251]. The authors studied the effects of the daily consumption of flavanone-rich orange juice (305 mg per dose) for 8 weeks in 37 healthy elderly people (mean age: 67 years). Cognitive function, mood and blood pressure were assessed at baseline and follow-up using standardized, validated tests. The results are positive for global cognitive function, which was significantly better after 8 weeks of the consumption of flavanone-rich juice than that after 8 weeks of the consumption of an orange-flavored but flavanone-free drink supplementation. There were no significant effects on the subjects’ mood or blood pressure (remember that they were healthy).

Another study with a different design [252] investigated whether the consumption of flavonoid-rich orange juice is associated with cognitive benefits in healthy middle-aged adults. Male subjects consumed 240 mL of orange juice (272 mg of flavonoids, including 220.5 mg of hesperidin) or a placebo sweetened drink. Cognitive function (with eight different tests) and subjective mood were assessed at baseline (before beverage consumption) and 2 and 6 h after consumption. The results showed that the performance in the tests of executive function and psychomotor speed was significantly better after the flavonoid-rich drink compared to that after the placebo. Curiously, the test that gave the best result in terms of the difference from the placebo was that of the “Single tip finger”, which involved pressing the “2” key as many times as possible with the index finger for 10 s. The same group performed another study [253] to investigate whether the consumption of juice rich in flavanone (500 mL of orange juice) is associated with acute cognitive benefits and changes in the cerebral circulation in young subjects (between 18 and 30 years old) undergoing cognitive tests. The drink intake was associated with a significant increase in regional perfusion in the right inferior and middle frontal gyrus at 2 h compared to the baseline and control drink. Additionally, the flavanone-rich beverage was associated with a significantly improved performance in a cognitive test (digit symbol replacement), but no effects on other cognitive or behavioral tests were observed. The authors suggest that further studies are needed to establish the mechanisms of the improved behavioral outcomes after citrus juice ingestion.

A basic epidemiological survey [254] collected data on the daily citrus intake (classified as </=2, 3–4 times/week or almost daily) and the consumption of other foods in a wide cohort of Japanese citizens from 2006 to 2012. Data on dementia incidence were retrieved from a Care Insurance Database. Among participants, the 5.7-year incidence of dementia was 8.6%. Compared with participants who consumed citrus </=2 times/week, the multivariate HR (hazard ratio, i.e., the risk related to those with low baseline citrus intake) for dementia among those who ate citrus 3–4 times/week and almost daily was 0.92 (95% CI 0.80–1.07) and 0.86 (95% CI 0.73–1.01), respectively, with a favorable trend (*p* = 0.065). Excluding patients whose dementia occurred in the first two years of follow-up, the multivariate HR was 1.00 (reference) for </=2 times/week, 0.82 (95% CI 0.69–0.98) for 3–4 times/week and 0.77 (95% CI 0.64–0.93) for nearly every day (*p* = 0.006). The results suggest that frequent citrus consumption was associated with a lower risk of incident dementia, even after adjusting for possible confounders.

Another epidemiological study explored the association between dietary flavonoid intake and cognitive health in a group of 808 Italian adults [255]. A significant inverse association was found between the increased dietary intake of flavonoids (catechins, anthocyanins and flavonols) and impaired cognition. Among several individual polyphenols, quercetin was most associated with cognitive health.

Holland et al. [256] investigated the association between doses and types of flavonoids (kaempferol, quercetin, myricetin, isorhamnetin) and cognitive performance over time in the elderly (age 60–100 years) among a cohort of 961 community-dwelling Chicagoans followed for 7 years. They filled out a yearly questionnaire about diet, lifestyle and memory tests such as remembering words and numbers and putting them in the correct order. The results demonstrated that people with the highest dietary intake of total flavonols showed a slower decline in cognition, even after adjusting for age, gender, education, physical activity and smoking. Analyses of the individual components of the diet showed that, of the four studied, quercetin was associated with a slower and more statistically significant cognitive decline. Myricetin and isorhamnetin were not associated with global cognition changes due to older age.

The effects of polyphenols on experimental AD models are summarized in Table 4. In this case too, most of the studies conducted so far have used models on in vitro neurons or rodents, and we present some characteristic examples of the experiments carried out in this field below.

Experimental evidence supports the hypothesis that some flavonoids may be beneficial in AD models, in part by supporting antioxidant mechanisms, in part by interfering with the generation and assembly of Aβ peptides in neurotoxic oligomeric aggregates and also by reducing the aggregation of tau [280]. Some mechanisms include the interaction with important signaling pathways in the brain such as phosphatidylinositol 3-kinase/Akt and mitogen-activated protein kinase pathways that regulate transcription factors and gene expression. Other processes include the interruption of Aβ aggregation and alterations in amyloid precursor protein processing through beta-secretase inhibition and/or alpha-secretase activation and the inhibition of the activation of cyclin-dependent kinase-5 and glycogen synthase kinase-3beta, preventing the abnormal phosphorylation of tau. The interaction of flavonoids with different signaling pathways highlights their therapeutic potential to prevent the onset and progression of AD and to promote cognitive performance.

A study investigated the effect of quercetin on normal laboratory mice, with negative results [281]. Laboratory mice were fed control and quercetin-enriched diets (2 mg/g diet) for 6 weeks. The supplementation significantly increased the plasma levels of quercetin and its methylated metabolite isorhamnetin, but the levels of alpha and beta-secretase and antioxidant enzymes in the cortex remained unchanged. This negative result suggests that the determining factors for the results are the dose, the timing and, above all, the model used (normal mouse or AD model). The results of the models used most often for studying the preventive and/or therapeutic effects of flavonoids in AD are reported below.

### 7.1. AD Transgenic Animal Models

AD-like neuronal degeneration spontaneously develops in the APP/PS1 transgenic mouse model. These laboratory mice are considered a model for early-onset AD because they express a chimeric human amyloid precursor protein and a mutant human presenilin, both of which target CNS neurons. In the study by Wang et al. [257], after 16 weeks of treatment with hesperidin, although there was no obvious change in the beta amyloid deposition in the treated group, the authors found a reduction in learning and memory deficits, an improvement in locomotor activity and increased antioxidant defense. In addition, the phosphorylation of glycogen synthase kinase-3beta (an enzyme whose dysregulation is associated with an increased susceptibility to bipolar disorders) was significantly increased in the hesperidin group.

Hesperidin significantly inhibited the inflammatory process and reduced the production of APP by the deposition of Aβ peptides in the hippocampus and cortex of the mouse brain with the APP/PS1 animal AD model [258]. After a relatively short-term treatment of 10 days, hesperidin significantly reversed the deficits in nesting ability and social interaction. Furthermore, effective anti-inflammatory effects of hesperidin have been confirmed in vitro. The authors suggested that hesperidin could be a potential candidate for the treatment of AD or even other neurodegenerative diseases.

Hong et al. [263] also demonstrated neuroprotective effects of hesperidin administration in transgenic APP/PS1 mice. The treatment with hesperidin attenuated cognitive impairment and suppressed oxidative stress by reducing the levels of lipid peroxides and by increasing the activity of heme-oxygenase, SOD, catalase and GSH peroxidase [263]. The treatment also inhibited inflammation, decreased TNF-alpha, C-reactive protein and MCP-1 levels, reduced NF-kappaB activity and increased Nrf2 protein. Remember that substances with the ability to stimulate the Nrf2 factor are considered potential remedies against AD [282].

Treatment with the flavonols quercetin and kaempferol in APP/PS1 mice at the age of 8 months for 4 months significantly reduced intracellular Aβ levels and improved BDNF expression [268]. These changes are related to improvements in cognitive parameters. Other authors found that, in APP/PS1 mice, a quercetin-enriched diet during the early-middle phase of the AD pathological developmental period improves cognitive dysfunction, and the protective effect was mainly related to decreased astrogliosis and increased Aβ clearance [283]. In the study by Wang et al. [272], three-month-old APP/PS1 transgenic mice were treated with quercetin for 16 weeks, observing beneficial effects, including a reduction in learning and memory deficits, a reduction in scattered senile plaques and an improvement in mitochondrial dysfunction, as evidenced by the restoration of the membrane potential and the normalization of reactive oxygen species and ATP levels in the mitochondria isolated from the hippocampus. Furthermore, AMP-activated protein kinase activity increased significantly in the quercetin-treated group.

Regitz et al. used the model of the nematode *Caenorhabditis elegans* CL2006, which expresses human Aβ and responds to Aβ aggregation with paralysis [271]. Quercetin in doses of 10 µMol/L and above dose-dependently decreased the amount of aggregated protein in the solution and also decreased paralysis. Quercetin enhanced Aβ degradation by promoting proteasomal activity and the macroautophagy pathway, which is the main mechanism for disposing damaged organelles or unused proteins by means of lysosomal enzymes. On the model of transgenic *Caenorhabditis elegans* that accumulates Aβ, a decrease in metallothionein, an inducible protein that detoxifies cells from metals such as copper and iron that can catalyze oxidative stress, has been noted [276]. These authors demonstrated that adding quercetin to the culture liquid of the organisms increases metallothionein, slows down the development of paralysis and extends the lifespan. The ability of quercetin to stimulate metallothionein expression through the Nrf2/ARE system was also demonstrated in liver cells subjected to metal intoxication [284].

The neuroprotective effect of quercetin administration was evaluated on mice of the triple transgenic AD model (i.e., with three mutations associated with hereditary AD, namely, APP-Swedish, MAPT-P301L and PSEN1-M146V) aged 21–24 months (for mice, this is an advanced age) [274]. The data show that quercetin decreases extracellular beta-amyloidosis, tau protein formation, chronic inflammation in the hippocampus and amygdala (astrogliosis and microgliosis). In addition, a significant reduction in BACE1-mediated APP cleavage (a key enzyme of amyloid plaque formation [285]) was observed. This treatment induced an improved performance in learning and spatial memory tasks and increased risk assessment behavior based on the elevated maze test.

Mouse APP23 overexpresses human APP with the Swedish mutation (KM670/671NL) in neurons and produces APP-derived extracellular Aβ plaques and intracellular Aβ aggregates. In these mice, the expression levels of transcription factor 4 (ATF4) for presenilin 1 and gamma-secretase are increased [275]. By feeding the mice a quercetin-rich diet, presenilin 1 expression and Aβ secretion were decreased, as was ATF4 expression. After long-term quercetin feeding, memory impairment in APP23 mice was delayed.

### 7.2. Direct Pathological Effects of Aβ

The injection of the Aβ(1–42) protein induces neuronal degeneration both in vivo (mouse) and in vitro. The results obtained by Ikram et al. [70] indicated that hesperetin reduced Aβ pathology and significantly attenuated oxidative stress, as assessed by Nrf2/HO-1 and LPO and ROS expression, in the hippocampus and cortex of mice and in HT22 cells in vitro. The authors emphasized that the result of hesperetin treatment is a multipotent effect, as it involves the inhibition of oxidative stress, neuroinflammation, death by apoptosis and cognitive consolidation, concluding that this flavanone may be a promising therapeutic agent for AD-like neurological disorders. Additionally, hesperetin improved synaptic integrity, cognition and memory processes.

In the experiments by Liu et al. [270], quercetin and rutin were administered to laboratory mice for 8 days after the intracerebral injection of Aβ(25–35), which causes memory loss. The results demonstrated that quercetin treatment improved learning and memory abilities. Simultaneously, a reduction in neurovascular oxidation, an enhancement of cholinergic activity, the regulation of the ERK/CREB/BDNF pathway and the inactivation of the RAGE-mediated pathway were observed in the temporal lobe cortex. The receptor for advanced glycation end products (RAGE) is a receptor of the immunoglobulin superfamily that plays various important roles in physiological and pathological conditions. Compelling evidence suggests that RAGE acts as both an inflammatory intermediary and a critical inducer of oxidative stress, underlying the pathophysiological changes that drive the AD process [286].

### 7.3. Aluminum

In addition to what has already been reported in the chapter on neurotoxicity, there have been some experimental investigations on the effect of hesperidin on aluminum-induced neuropathology models. These are reported in this chapter because they are closely related to AD, for which this metal is considered one of the possible factors, though certainly not the only one [229,287,288,289]. Aluminum has been reported to cause the apoptotic neuronal loss associated with oxidative stress in the brain [290]. Evidence for the neurotoxicity of Al is described in various studies and suggests that Al may potentiate oxidative stress, reduced glutathione depletion, decreased mitochondrial integrity and increased proinflammatory cytokine production both in the brain and peripherally [229].

In a rat AD model exposed to aluminum chloride (AlCl3), hippocampal lesions and behavioral disturbances were markedly attenuated by hesperidin, reducing Aβ(1–40) levels by inhibiting the β- and γ-secretases responsible for the protein cleavage amyloid precursor [260]. At the same time, hesperidin attenuated the losses of reduced glutathione caused by aluminum and supported the activity of antioxidant enzymes. Furthermore, hesperidin reversed the memory loss caused by aluminum intoxication in rats through the attenuation of acetylcholine esterase activity and the expression of Aβ biosynthesis-related markers [261]. In addition, the intraperitoneal injection of AlCl(3) significantly elevated the expressions of the insulin degrading enzyme, phosphoTau protein (typical marker of AD), inflammatory markers such as glial fibrillary acid protein, NF-kappaB, cyclooxygenase- 2, IL-1beta, IL-4, IL-6, TNF-alpha and inducible nitric oxide synthase and apoptotic markers including cytochrome c and others in the hippocampus and cortex [71]. However, the co-administration of hesperidin significantly ameliorated these aluminum-induced pathological changes. Behavioral studies have also supported these findings.

Regarding the aluminum AD model, some researchers studied the effect of hesperidin in comparison with donepezil and limonoids [266]. Donepezil is an AChE inhibitor which, after the intake, allows for a slowdown in the degradation of acetylcholine. Limonoids are natural organic substances present in citrus seeds such as limonin, nomilin and nomilinic acid. Donepezil, hesperidin and limonoids were used to treat rats for 2 weeks before the concomitant administration of AlCl(3) for the following 3 weeks. Treatment with both donepezil and hesperidin, or high doses of limonoids alone (100 mg/kg), produced an improvement in the psychological status, as observed by a significant increase in the frequency of walking compared to the group treated with aluminum. Regarding the levels of acetylcholine esterase and Aβ, the effect was positive in terms of bringing the parameters back to the levels of the group not treated with aluminum.

### 7.4. Other AD Models

In addition to causing depression (see Section 5.4), intracerebroventricular treatment with STZ mimics some elements of AD pathology. In this albino rat model of sporadic dementia type AD, hesperidin showed a significant neuroprotective effect by improving memory, spatial learning, motor functions and cholinergic dysfunction [259,264]. Hesperidin has shown anti-inflammatory activity characterized by the inhibition of COX2, iNOS and NF-kappaB, inhibiting the activity of AChE and lipid peroxidation by reducing the levels of thiobarbituric acid reactive substances (TBARS) and increasing the level of gangliosides. The treatments showed improvement in recognition memory symptoms and increased the activity of antioxidant enzymes (SOD, glutathione GPx, GRx and CAT) and GSH levels and reduced malondialdehyde in the hippocampal area. In the same model of AD- and STZ-induced memory impairment, quercetin ameliorated spatial memory impairment and oxidative stress, and the positive effect was synergistic with that of moderate physical exercise (treadmill for one hour per day for 60 days) [277]. This result is interesting because it corroborates the hypothesis that diet and lifestyle are both important for the prevention of neurodegenerative diseases.

The administration of scopolamine (an anticholinergic drug that interferes with the effects of the neurotransmitter acetylcholine) induces AD-mimicking behavior in rats, especially with regard to memory deficits. The concomitant administration of hesperidin and donepezil (a drug used to counter the symptoms of dementia) showed significant neuroprotection in the improvement of dementia and cognitive impairment in this animal model [262]. Another similar but shorter-term animal model study investigated the effect of hesperetin on scopolamine-induced memory impairment in mice [265]. This study demonstrated that hesperetin enhanced nonspatial and spatial learning and reduced memory impairment by scopolamine, possibly due to improvements in antioxidant defense, cholinergic signaling and neurotrophic factors in the brain. Other authors confirmed the neuroprotective effect of hesperidin in the mouse AD model by treatment with scopolamine [267]. This latter investigation showed that memory impairment was associated with a concomitant increase in serum TNF-alpha and IL-1beta, while IL-10 was significantly decreased. In addition, there was a pathological increase in Aβ(1–42), AChE and malondialdehyde (plasma indicator of oxidative stress), along with a decrease in reduced glutathione in the hippocampal and prefrontal homogenate. The 4-week pre-treatment with hesperidin compensated for spatial memory deficits, redox imbalance, Aβ(1–42) and AChE activity, and it preserved the histological architecture. The authors suggest the possibility of its future implementation as a prophylactic remedy against AD in humans.

### 7.5. In Vitro and Bioinformatics Studies

An in vitro study investigates whether hesperetin can counteract the AD-like pathophysiological changes induced by advanced glycation (AGE) in SH-SY5Y neuronal cells [291]. The term AGE refers to a series of chemical compounds produced when sugars combine with proteins or fats. Cells pre-treated with hesperetin (40 µmol/L) before contact with AGE (200 µg/mL) showed improved viability, reduced reactive oxygen species overproduction and normalized superoxide dismutase, glutathione peroxidase and catalase. The upregulation of amyloid precursor protein, accompanied by increased Aβ production, caused by AGEs, was reversed by hesperetin. Hesperetin-pre-treated cells had less apoptotic DNA fragmentation, less Bax expression and also decreased caspase-12/-9/-3 activity, indicating that it inhibits biochemical stress-mediated neuronal apoptosis.

Given the importance of cholinergic transmission and its decay in AD, some authors have tested the effects of quercetin on the key enzymes involved in this pathology. An initial study [292] demonstrated that quercetin inhibits AChE, with an IC50 value of 19.8 μMol/L. The inhibitory activities of various flavonoids (genistein, biocanin A, naringin, apigenin, quercetin, luteolin-7-O-rutinoside, kaempferol-3-O-galactoside, diosmin, silibinin and silymarin) were also tested in vitro against AChE and BChE (butyrylcholinesterase, another enzyme that degrades acetylcholine) [293]. Of the various polyphenols tested (they did not test hesperidin), only quercetin showed substantial inhibition (76.2%) against AChE, while genistein, luteolin-7-O-rutinoside and silibinin exerted moderate inhibition on BChE. The physicochemical properties of quercetin in binding to AChE were also investigated using bioinformatics tools [294] with “molecular docking” software. Quercetin showed an AChE molecular docking score similar to or higher than that of conventional drugs such as donepezil. The strong inhibition of AChE by quercetin has also been seen by others [295,296].

Another study tried to evaluate the effect of quercetin on synapse loss in an in vitro model based on hippocampal neurons [297]. This model is particularly interesting because synapse loss has been related to dementia in AD as an early event during disease progression. Of the compounds contained in Gingko biloba extract, bilobalide (a main constituent of the terpenoids found in Ginkgo leaves) and quercetin increased cell proliferation in hippocampal neurons in a dose-dependent manner. Bilobalide and quercetin also enhanced the phosphorylation of cyclic AMP response element binding protein (CREB) in these cells and elevated pCREB and BDNF levels in the brains of mice. The immunofluorescence staining of synaptic markers shows prominent dendritic processes in hippocampal neurons treated with quercetin or bilobalide. Furthermore, both constituents counteracted the synaptic loss and CREB phosphorylation induced by Aβ oligomers.

Bioinformatic analysis combined with an experimental verification strategy was used to identify biomarkers and targets of quercetin for the diagnosis and treatment of AD [298]. Differentially expressed genes in the AD brain were identified by a microarray, and quercetin was used to screen for target genes in the neuronal cell line HT-22 (commonly used for glutamate- and amyloid-induced toxicity studies). The authors found that the biological activity of quercetin affects biological processes such as the negative regulation of the apoptotic process and neuronal migration. The results suggested that the expression of MAPT, PIK3R1, CASP8 and DAPK1 genes was significantly increased after Aβ(1–42) treatment, but the effect was counteracted by quercetin treatment. Furthermore, the important genes were mainly distributed in the PI3K-AKT signaling pathway. This pathway has been proposed by others as well [174], and PI3K-AKT inhibition has also been suggested in the modulation of human basophils by quercetin [299,300]. Other analyses of the molecular interactions of quercetin with the reconstruction of protein–protein networks and docking simulations identified six targets [301]: AKT1, JUN (an oncogene), MAPK (mitogen-activated protein kinase), TNF-alpha, VEGFA (vascular endothelial growth factor) and EGFR (epidermal growth factor receptor).

There are many other laboratory investigations that try to explain the mechanisms of the protective effects of quercetin. After an extensive screening of medicinal plants, with liquid chromatography mass spectrometry studies, it was demonstrated that quercetin and five other flavonoids (datiscetin, kaempferol, morin, robinetin and myricitrin) are capable of inhibiting the aggregation of the Aβ(1–42) protein [302], confirming other evidence of this phenomenon [303]. Another interesting mechanism invoked in order to explain the positive effect of polyphenols (quercetin, resveratrol, kaempferol and epigallocatechin gallate) is the inhibition of oxidative stress caused by apolipoprotein E4 (apoE4), some variants of which (L28P) are related to the development of AD. In fact, these molecules prevented the redox state changes induced by Aβ uptake in neuronal cells treated with apoE4-L28P in vitro [304].

In addition, there are biological mechanisms, such as the regulation of sirtuin [305,306], the expression of thioredoxin-interacting protein (TXNIP) [54], the inhibition of the formation of fibrils of Aβ proteins, the inhibition of MAO, and the inhibition of tyrosinase [247].

Collectively, the results of these studies suggest that hesperidin and quercetin could be useful complementary agents in the prevention and treatment of oxidative stress and toxicant-induced neurodegenerative diseases, including AD. All of this evidence is particularly interesting in light of the fact that hesperidin can cross the blood–brain barrier, also thanks to its lipophilic properties [307].

## 8. Parkinson’s Disease

Parkinson’s disease (PD) is a neurological condition with the selective progressive degeneration of dopaminergic neurons. The disease causes involuntary or uncontrollable movements, such as tremors, stiffness and difficulty with balance and coordination. Symptoms usually start gradually and get worse over time. As the disease progresses, people may have difficulty walking and talking. The name is related to James Parkinson, a 19th century London surgeon, who first described most of the symptoms of the disease. Routine therapies are symptomatic and palliative. A population-based study of Medicare beneficiaries in the United States found a mean prevalence of 1.6% for PD among people aged 65 and older [308].

The most prominent signs and symptoms of PD occur when nerve cells in the basal ganglia, an area of the brain that controls movement, deteriorate and/or die. Normally, these nerve cells, or neurons, produce an important brain chemical known as dopamine. When neurons die or deteriorate, they produce less dopamine, which causes the movement problems associated with the disease. Scientists still do not know what causes the death of neurons, but it is certain that pathological phenomena linked to oxidative stress and chronic inflammation are involved in this case too, which are contrasted by the use of food substances such as polyphenols [309].

The causes of the disease are somewhat obscure, even if some genetic defects are known, which lead to the accumulation of substances such as α-synuclein in the form of inclusion bodies. α-synuclein is a protein that regulates synaptic vesicle trafficking and neurotransmitter release; it is classically considered to be a soluble protein, but it can form insoluble protein aggregates inside neurons in various neurological pathologies (Lewy bodies). The cells most often affected are those of the “substantia nigra” of the brain and, precisely, those that implement a dopaminergic nerve transmission, controlling the precision of movements. Mutations in the Leucine-rich repeat kinase 2 (LRRK2) gene, which encodes protein kinase enzyme activity, have been identified in the familial forms. Mutations of the LRRK2 gene increase the enzymatic activity of LRRK2, and this phosphorylation induces a long-term aberrant accumulation of some proteins [310]. LRRK2 dysfunction has been reported to affect the accumulation of α-synuclein, a major stimulant of microglial activation. Microglial activation is thought to contribute to neuroinflammation and neuronal death in PD [311].

The pathogenic role of MAO is also found in PD, both because it can catalyze radical reactions and because it degrades dopamine, which has neurotransmitter functions. With aging, MAO and iron levels in the brain increase, which leads to an increase in Fenton reaction components and macromolecule damage. Furthermore, Fe^3+^ from the Fenton reaction directly induces the expression and aggregation of ɑ-synuclein [240]. Therefore, the MAO inhibition by the flavonoid utilization or the Fe^2+^ ion removal by an iron chelator are two approaches with the same target in PD patients, because they increase dopamine levels and decrease oxidative stress at the same time [312].

In recent years, a large number of molecules of plant origin have been reported as having significant MAO inhibition potential [313,314]. Differently substituted flavonoids have been prepared and studied as MAO-A and MAO-B inhibitors [146,183,184,185]. In addition to what has already been seen with hesperidin in the case of AD, quercetin, myricetin and chrysin showed potent inhibitory activity directed towards MAO-A, while genistein inhibited MAO-B more efficiently [315].

Some features of the pathology of PD are represented schematically in Figure 7.

Experimental cell and animal models of PD report the protective effects of the flavonoids considered, which are briefly described here and summarized in Table 5.

### 8.1. Hydroxydopamine-Induced PD

The first experimental work on laboratory animals was published in 2014 by Antunes and collaborators [319]. The purpose of this study was to evaluate the role of hesperidin on a 6-hydroxydopamine (6-OHDA)-induced mouse PD model. 6-OHDA is a neurotoxin used by researchers to selectively kill dopaminergic and noradrenergic neurons because it enters neurons using monoamine transporters. It is especially used to induce parkinsonism in laboratory animals in order to test potentially useful PD drugs in humans. Aged mice were treated with hesperidin for 28 days after an intracerebroventricular injection of 6-OHDA. This study demonstrated that hesperidin treatment was effective in preventing memory impairment, as well as depressive-like behavior. Hesperidin attenuated the 6-OHDA-induced reduction in glutathione peroxidase and catalase activity, dopamine levels and its metabolites in the striatum. The same authors [320,321] deepened the knowledge of the mechanisms underlying the neuroprotective effects of hesperidin, demonstrating that the treatment improved anxious and depressive behavior in mice with 6-OHDA lesions. It also attenuated the striatal levels of proinflammatory cytokines such as TNF-alpha, IFN-gamma, IL-1beta, IL-2 and IL-6 and increased the levels of neurotrophic factors, including neurotrophin-3, BNDF and NGF. Hesperidin treatment was also able to increase the striatal levels of dopamine and its metabolite 3,4-dihydroxyphenylacetic acid. In conclusion, this study indicated that hesperidin exerts an anxiolytic-like and antidepressant-like effect through the modulation of cytokine production, neurotrophic factor levels and dopaminergic innervation in the striatum. Taken together, the results indicate that hesperidin mitigates the degeneration of dopaminergic neurons in the central nervous system by preventing mitochondrial dysfunction and modulating apoptotic pathways, thereby ameliorating behavioral alterations.

Quercetin was also evaluated on the 6-OHDA induced model of PD [326]. PD was induced by a single intracisternal injection of 6-OHDA into male Sprague–Dawley rats. Treatment with quercetin for 14 consecutive days markedly increased the levels of striatal dopamine and antioxidant enzymes compared with similar measurements in the group treated with 6-OHDA alone. The efficacy of quercetin in the 6-OHDA-induced model of PD has also been proven by others [329]. The administration of quercetin (10 and 25 mg/kg) prevented memory impairment, increased antioxidant enzymes (superoxide dismutase and catalase) and total glutathione and reduced the level of malondialdehyde (MDA) in the hippocampus area. A preparation of quercetin in the form of nanocrystals proved to be slightly more effective, probably due to the greater bioavailability. Finally, a meticulous study by Chinese authors tested quercetin on both cells and experimental animals in PD models with the administration of 6-OHDA [332]. In the 6-OHDA-treated PC12 neuronal line, the authors found that quercetin treatment improved mitochondrial functioning, reduced oxidative stress and decreased α-synuclein expression. In vivo results demonstrated that quercetin administration to rats alleviated 6-OHDA-induced progressive PD-like motor behavior, attenuated neuronal death and reduced mitochondrial damage and α-synuclein accumulation.

### 8.2. Rotenone-Induced PD

Rotenone, a widely used pesticide that inhibits mitochondrial complex I, has been used to simulate the pathobiology of PD both in vitro and in vivo. This model was used to study the neuroprotective effects of hesperidin against rotenone-induced apoptosis in human neuroblastoma cells in vitro [318]. The data indicate that hesperidin exerts its neuroprotective effect against rotenone due to its antioxidant potency, maintenance of mitochondrial function and antiapoptotic properties.

Quercetin also showed remarkable dose-dependent neuroprotective effects on rotenone-induced parkinsonism in rats, significantly reducing catalepsy and improving behavioral tests (rotarod) [330]. The flavonol reversibilized the decline in glutathione, SOD, catalase and serum iron and lipid peroxidation, synergizing with L-dopa. In silico molecular docking studies also demonstrated that potential drug targets for quercetin could be the aromatic L-amino acid decarboxylase and catechol-O-methyltransferase enzymes. The animal PD model induced by rotenone injections has also been used by others to study the effect of flavonoids in the pathogenesis of the disease [331]. The results showed that quercetin supplementation significantly reversed rotenone-induced motor and non-motor deficits (depression and cognitive deficits), improved antioxidant enzyme activities and attenuated neurotransmitter alterations. Pre-supplementation with quercetin produced more significant results than post-supplementation.

Josiah et al. investigated the neurobehavioral, brain redox stabilizing and neurochemical modulatory functions of catechin (tea flavonoid) and quercetin in rotenone-induced parkinsonism and the involvement of inflammation [333]. Both flavonoids attenuated striatal redox stress and neurochemical dysfunction and optimized dopamine metabolism disturbed by rotenone toxicity. While catechin administration produced a more pronounced attenuating effect on the IL-1beta, TNF-alpha and p53 genes, the attenuating effect of quercetin was more pronounced on the expressions of the NF-kappaB and IkappaKB genes compared to the rotenone-only group. Comparatively, quercetin demonstrated superior protection against rotenone neurotoxicity.

### 8.3. Ferroptosis and Neurodegeneration in PD

Ferroptosis has been linked to various pathophysiological conditions, such as hemorrhage, neurodegeneration, ischemia/reperfusion injury, diabetes and PD [334,335]. Elevated iron concentrations have been observed in various brain areas, including the cerebral cortex, hippocampus, cerebellum, amygdala and basal ganglia, in healthy older adults; it is most likely that these areas are involved in neurodegenerative diseases [240].

The neurodegenerative effects of iron overload lead to decreased brain functions, with demonstrable effects on dopamine levels, for which a model of PD induced by iron intake in Drosophila melanogaster (fruit fly) was developed [322]. The midges develop very quickly and therefore allow for laboratory studies within a short time. A negative correlation was found between the Fe levels in the fly’s head and survival, dopamine levels and antioxidant enzymes in the fly’s head. An important finding of the study is that adding hesperidin to the diet promotes a decrease in the Fe concentration in the head, restores the dopamine levels and cholinergic activity of flies and improves Fe-damaged motor function. Hesperidin also prevents oxidative stress and mitochondrial dysfunction.

As iron accumulation in the brain leads to the development of AD and PD, iron chelation therapy has been proposed to reduce the pathogenic effects. Male NMRI mice (a strain commonly used as experimental animals in biology, pharmacology and toxicology) were treated with iron dextran for 6 weeks and then divided into groups and treated with hesperidin, coumarin, desferal (a conventional iron chelator) or nothing for the next 4 weeks [312]. The results show that hesperidin and coumarin strongly chelate excess iron from serum and inhibit iron deposition from brain tissue. Catalase and SOD activity were reduced in the iron overload group, but in the hesperidin and coumarin group, the enzyme activity increased significantly.

It is also interesting that molecular simulations have demonstrated the ability of quercetin and naringenin to bind to human ferritin [336], which may be important in light of the fact that, as we have seen, iron participates in neuronal damage in various ways, even in AD (“ferroptosis”).

### 8.4. Transgenic Mouse

The “MitoPark” mouse model was developed as an animal PD model by inactivating TFAM (Transcription Factor A, Mitochondrial) in dopaminergic neurons. The MitoPark mouse summarizes several aspects of human PD, such as adulthood onset, the progressive degeneration of dopaminergic neurons and the response to levodopa. This preclinical model was used to test the potential neuroprotective effect of quercetin for the treatment of PD [328]. The oral administration of quercetin significantly reversed behavioral deficits, striatal dopamine depletion and neuronal cell loss. Quercetin also enhanced brain-derived neurotrophic factor expression and mitochondrial biogenesis. In summary, the authors suggest that the further exploration of quercetin as a promising neuroprotective agent for the treatment of PD may offer clinical benefits.

### 8.5. In Vitro Studies

An in vitro study on cultured cortical neurons [316] showed that peroxynitrite (a molecule that is part of the ROS; see Figure 2) generates 5-S-cysteinyl-dopamine and dihydrobenzothiazines (DHBT-1), following the reaction of dopamine with l-cysteine. The formation of 5-S-cysteinyl-dopamine and DHBT-1 in the presence of peroxynitrite induced significant neuronal damage. The pre-treatment of cortical neurons with pelargonidin, quercetin, hesperetin, caffeic acid, the 4’-O-Me derivatives of catechin and epicatechin resulted in concentration-dependent protection against 5-S-cysteinyl-dopamine-induced neurotoxicity. These data suggest that polyphenols may protect against the neuronal damage induced by endogenous neurotoxins that are relevant to PD etiology.

There is also a cellular model of PD based on the in vitro intoxication of cells with 6-OHDA. Considering the important role of calcium (Ca^2+^) in the cellular mechanisms of neurodegenerative diseases, a study aimed to investigate the possible effects of hesperidin on Ca^2+^ channels in models of PD neuronal lines [323]. The incubation of SH-SY5Y cells with hesperidin showed a protective effect, reduced biochemical markers of cell damage/death and balanced intracellular calcium homeostasis. With the same cellular model, hesperidin was found to significantly protect SH-SY5Y cells from 6-OHDA-induced toxicity, reducing oxidative stress biomarkers [324]. Ahn et al. investigated the neuroprotective role of quercetin against various neurotoxin toxins on PC12 cell lines [337]. Cells were pre-treated with quercetin and then with 6-OHDA, ammonium chloride (an autophagy inhibitor) and nocodazole (an aggression inhibitor). Cell viability was protected by lower concentrations (100 and 500 muMol/L) of quercetin but decreased at very high concentrations (1000 muMol/L).

Another in vitro study evaluated the effect of resveratrol and quercetin on the apoptotic cascade induced by the administration of the 1-methyl-4-phenylpyridinium ion (MPP(+)), a toxin mimicking PD-like damage, causing the selective degeneration of dopaminergic neurons [338]. The results show that pre-treatment for 3 h with resveratrol or quercetin before the administration of MPP(+) remarkably reduces the neuronal death induced by MPP(+). The same group confirmed the efficacy of quercetin in a subsequent investigation [339] conducted on a glial-neuronal system. Quercetin and sesamin (a polyphenol isolated from sesame oil) defend microglial cells against MPP(+)-induced increases in the levels of IL-6, IL-1beta and TNF-alpha. The result of the co-culture of microglial cells and neurons is also very interesting: quercetin and sesamin rescued PC12 neuronal cells from apoptotic death induced by the MPP(+) activation of microglial cells.

The study by Bahar and co-workers aimed to investigate the protective effect of quercetin on manganese (Mn)-induced neurotoxicity in the human neuroblastoma cell line SK-N-MC and the rat brain [327]. Mn is an essential trace element necessary for the development of the human body and acts as an enzyme cofactor or activator for various metabolic reactions, but excessive exposure, usually of occupational origin, can cause extrapyramidal symptoms called manganism, similar to PD. In the study cited [327], the treatment with Mn significantly reduced cell viability and increased lactate dehydrogenase release; these alterations were attenuated by the pre-treatment with quercetin. Meanwhile, the pre-treatment markedly downregulated NF-kappaB but upregulated the heme oxygenase-1 (HO-1) and Nrf2 proteins, as compared with the Mn-only group.

In a model of neuronal cells intoxicated with glutamate, the “scavenger” action of quercetin was protected by ascorbate (vitamin C), which prevents the pro-oxidant action of its quinone form that occurs during the prolonged exposure to oxidative stress [340]. The synergism between quercetin and ascorbate has also been highlighted by others [341].

Copper (Cu) is one such metal that serves as an essential cofactor for the activity of several enzymes, one of which is cytochrome c oxidase. The growing body of evidence suggests a substantial correlation between Cu overload and neurodegenerative disorders, including PD [342]. The authors cited the investigated quercetin as an alternative in combatting Cu-induced toxicity in SH-SY5Y human neuroblastoma cell lines. They observed that Cu increases intracellular ROS levels, triggers morphological alterations such as nuclei condensation, causes a mitochondrial membrane potential imbalance and ultimately induces apoptotic cell death. However, quercetin reversed these changes due to its antioxidant and anti-apoptotic properties by modulating autophagic pathways.

Quercetin and kaempferol are fundamental components of Suanzaoren decoction (SZRD) used in traditional Chinese medicine in the treatment of PD, particularly in the presence of sleep disorder [343]. The aforementioned study tried, using network pharmacology and “molecular docking” methods, to understand the path that starts from the herbs used and moves on to the components and, finally, to their biological targets. The results showed that the active compounds could bind to key targets of AKT1, IL6, MAPK1, TP53 and VEGFA. This study provides a theoretical basis for further studies on the material basis and molecular mechanism of such a decoction in the treatment of PD.

## 9. Effects on Aging Models

Flavonoids have positive effects on aging-related diseases [344], but few studies have explored whether they also exert independent antiaging effects [345]. On simple and unicellular models, senolytic effects of flavonoids have been observed, which prolong the biological life of senescent cells [346], the yeast *S. cerevisiae* [347], human keratinocytes [348,349], adipocytes [350,351], fibroblasts [352], umbilical vein endothelial cells [353] and the intervertebral disc (nucleus pulposus) [354]. A few experiments have observed an extension of the lifespan of the nematode *Caenorhabditis elegans* [355,356,357], Drosophila melanogaster [358] and mice [359]. In the latter case, the effect was accredited to the action on the intestinal microbiota. Although the antioxidant properties of flavonoids make an antiaging effect highly plausible, further studies are needed to clarify whether they really have positive effects in prolonging human life.

## 10. Synthesis and Conclusions

In view of the need for a better understanding of the action mechanisms of the various natural substances, this narrative review aimed to analyze the relevant literature in order to highlight the state of knowledge and the consequent efficacy of flavonoids in the field of the main nervous and neurodegenerative diseases. Collectively, the results confirm that the antidepressant and neuroprotective effects of flavonoids are part of a broad-spectrum effect on the nervous system, involving various molecular targets that are part of the major vicious cycle of neuropathology, composed of oxidative stress, cytotoxicity and inflammation pertaining to the biochemical steps implicated in neuropathology. In an attempt to summarize a lot of information, three general and fundamental mechanisms of action can be considered: the inhibition of oxidative stress, the protection of cellular structure and function and the modulation of inflammation.

### 10.1. Inhibition of Oxidative Stress

The powerful antioxidant actions of flavonoids have been demonstrated in various ways, specifically with the inhibition of enzymes that produce ROS such as xanthine oxidase [24,272,360] and MAO [146,183,184,185,313], the reduction in nitric oxide production [145,153,165,175,194,259,316] and the increase in the resistance to ischemia [204,213,227,361,362,363,364,365] and to the many chemicals known to trigger oxidative stress [32,142,205,206,207,208,209,210,214,215,216,217,218,219,220,222,224,226,237,238,239,240,273,336]. In multiple experimental in vitro and in vivo models, a decrease in lipoperoxidation markers was observed [70,150,163,206,218,219,237,238,239,264,324], along with the remarkable ability to activate the Nrf2/ARE system and therefore the expression of antioxidant defense enzymes [70,156,157,159,172,173,176,204,218,221,224,226,263,319,327,366].

### 10.2. Protection of Cellular Structure and Function

Experimental studies have demonstrated in various ways that flavonoids possess powerful actions that counteract cytotoxicity and preserve the normal function of neurons subjected to various types of pathogenic insults. For example, MAO inhibition in models of stress of various types, depression and PD [146,183,184,185,313] has a dopamine-preserving effect, with a potential improvement of neurotransmission in those compromised conditions. Furthermore, protection from excitotoxic damage by glutamate or the excessive stimulation of NMDA receptors has been demonstrated [24,41,165,340,367,368,369], as well as the protection of dopamine levels [162,322,328,333], the interaction with the serotonin system [144,146,154,370,371,372,373] and the modulation of the neuroendocrine axis [164,183]. In AD models, the flavonoids considered have demonstrated the ability to inhibit secretases, enzymes involved in the formation of oligomers and to prevent the accumulation of Aβ aggregates [260,268,271,272,283,302,374,375]. They also inhibited AChE activity, and this could preserve the cholinergic transmission so often compromised in this type of disease [259,266,267,273,292,293,294,295,296]. In different disease models, there is a lot of evidence that flavonoids stopped the biochemical and metabolic processes of apoptosis [16,53,169,214,224,240,261,291,298,318,376] and protected or promoted cell viability via the CREB/BDNF [145,150,160,174,178,207,268,270,297,320,328,377].

### 10.3. Inflammation Modulation

As we have seen, neuroinflammation is closely intertwined with oxidative stress and cytotoxicity, so the beneficial effects of flavonoids are also expressed as anti-inflammatory effects [4]. The most frequent evidence of this efficacy reported in the literature concerns the inhibition of the formation of cytokines (especially TNF-alpha and IL-1beta) by microglia and astrocytes [157,163,165,177,207,263,267,279,378,379] through the modulation of NF-kappaB [71,151,175,194,214,216,259,320,333,380,381,382,383,384] and NLRP3/inflammasome [54,175,385,386,387,388,389,390].

It should be noted that the limited clinical studies on these natural products determine an important constraint on clinical use in human diseases, which deserves further research. In the same way as other supplements and phytotherapeutic or nutraceutical products, “evidence-based” clinical research is still scarce, which is also due to obvious cost problems, since these are natural substances that are not patentable, and clinical trials in chronic diseases have to last for years. The bioavailability, effective dose, tolerability and possible synergisms and antagonisms with other drugs often used in elderly people have yet to be better investigated. In light of preclinical studies and initial human clinical evidence, these aspects deserve to be explored by performing well-designed clinical trials in patients suffering from different forms of neurodegenerative disorders.

The extent of the results presented here is such that it constitutes strong proof of the efficacy of hesperidin and quercetin in several experimental models of the pathologies mentioned and of their multiple mechanisms of action. This should make any further studies on experimental animals with the same or similar models less and less necessary. Even if the studies referred to here are normally approved by ethical committees for animal experimentation, the models of neurological pathologies are particularly challenging and are ethically justifiable only in the absence of alternative models and if they are essential to clarifying still unknown and truly unexplored issues. Research in these fields could proceed through adequate clinical cohort studies, even randomized ones, if possible, comparing groups of human subjects whose diets have types of supplements added to them with those on a normal diet. Certainly, this type of study on chronic diseases is very long and demanding, as they are subject to various types of confounders, but the importance of the matter for the future of humanity amply justifies it. Since these are low-cost and non-patentable natural substances that are of interest to all of society, clinical studies such as those outlined here should have support from state public bodies or international health agencies.

In conclusion, there is extensive literature suggesting that hesperidin and quercetin are flavonoids with promising ability to prevent and treat even serious neurological diseases. Many studies describe the multiple mechanisms of the neuroprotective activity of flavonoids and suggest that a diet rich in quercetin and hesperidin, either as a diet rich in fruits and vegetables or as a food additive, can help cope with the stress caused by many pathogenic factors. It is suggested that the integration of these molecules, typical of the Mediterranean diet, improves the levels of neurotransmitters by mitigating oxidative stress by increasing antioxidant enzyme activity and therefore improving motor activity and cognitive functions and reducing depressive behavior. In any case, intervention in the diet or the additional use of supplements possibly associated with (and not as an alternative to) therapies of a certain efficacy must be carried out according to an evaluation by the doctor, based on biological plausibility and a series of individual assessments, particularly in cases in which challenging diseases such as neurological diseases are often faced.

## Figures and Tables

**Figure 1 antioxidants-12-00280-f001:**
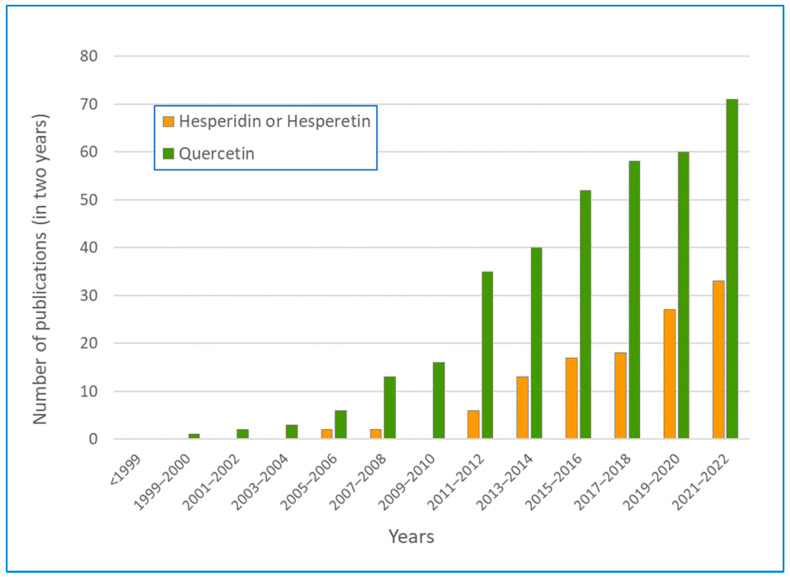
Number of publications in PubMed extracted with a keyword for the title (“Hesperidin or Hesperetin” or “Quercetin”) and for the abstracts (“Neuroprotective”).

**Figure 2 antioxidants-12-00280-f002:**
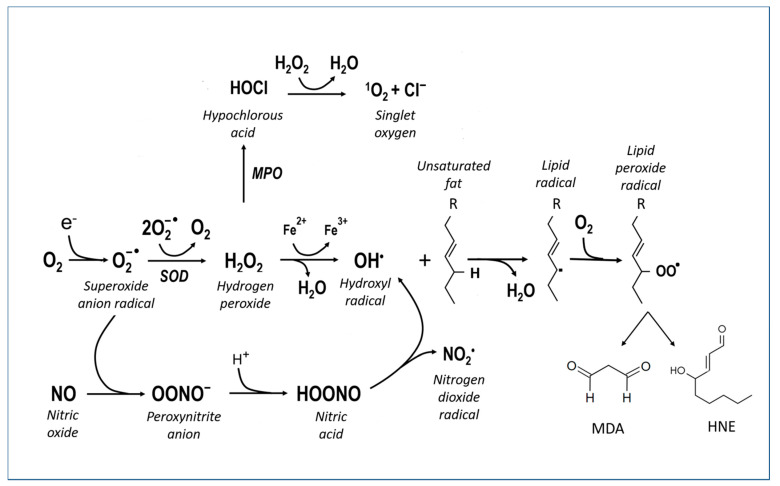
ROS and correlated chain reactions. MPO: myeloperoxidase; SOD: superoxide dismutase; MDA: Malondialdehyde; HNE: 4-hydroxynonenal.

**Figure 3 antioxidants-12-00280-f003:**
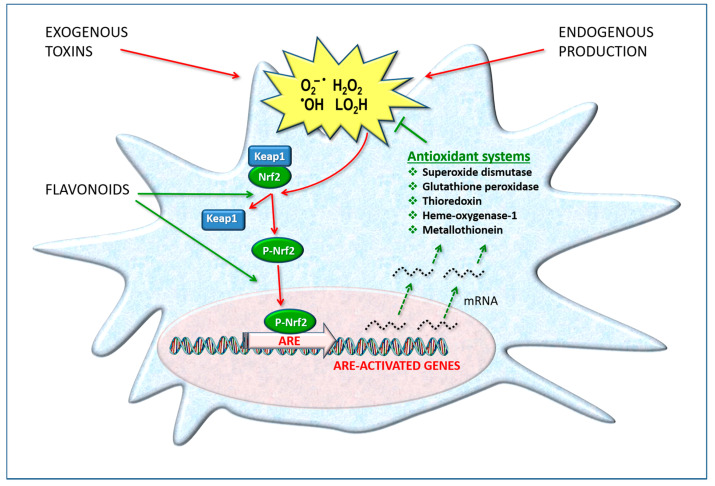
Diagram of the function of the Nrf2/ARE system. Nrf2: Nuclear factor erythroid 2-related factor 2; ARE: antioxidant response elements; Keap1: Kelch ECH associating protein.

**Figure 4 antioxidants-12-00280-f004:**
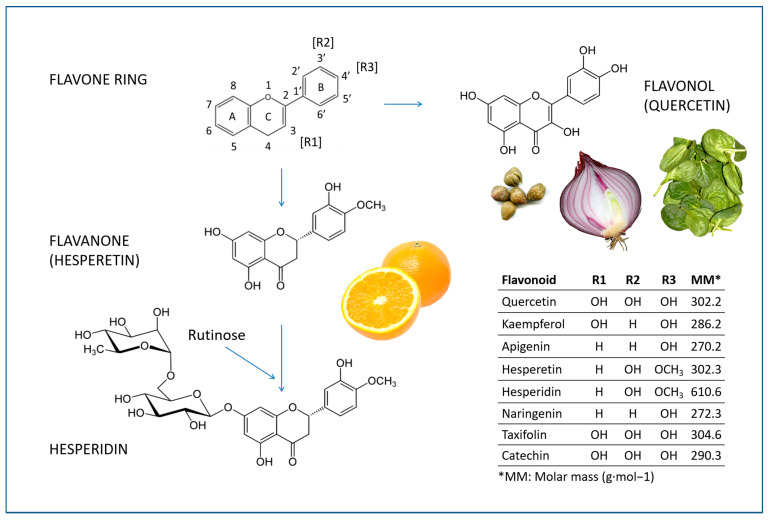
Structure of some flavonoids mentioned in the text. Hesperetin and hesperidin are characteristic components of citrus fruits, quercetin is present in many vegetables, among which capers, onions and spinach are very rich.

**Figure 5 antioxidants-12-00280-f005:**
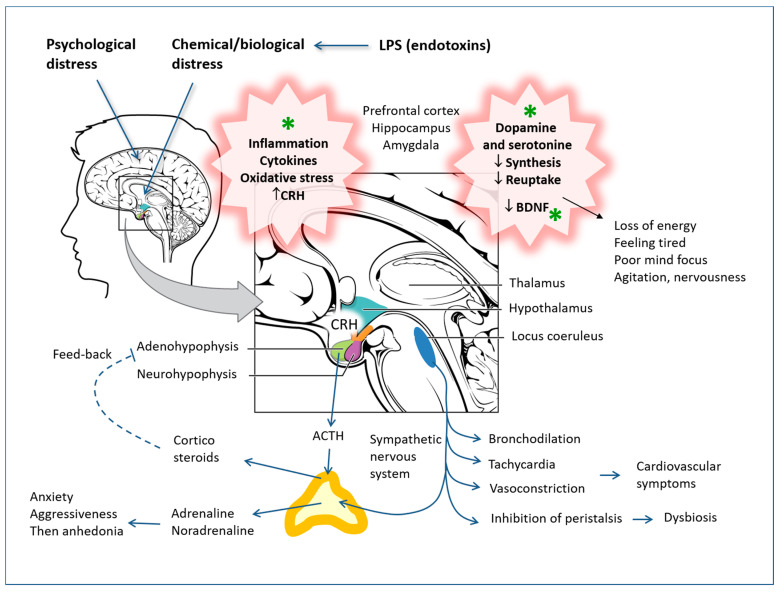
Some pathogenetic mechanisms of depression described in the text. CRH: corticotropin-releasing hormone; ACTH: adrenocorticotropic hormone; BDNF: brain neurotrophic factor. Asterisks indicate the proposed action points of the flavonoids that are described in the text.

**Figure 6 antioxidants-12-00280-f006:**
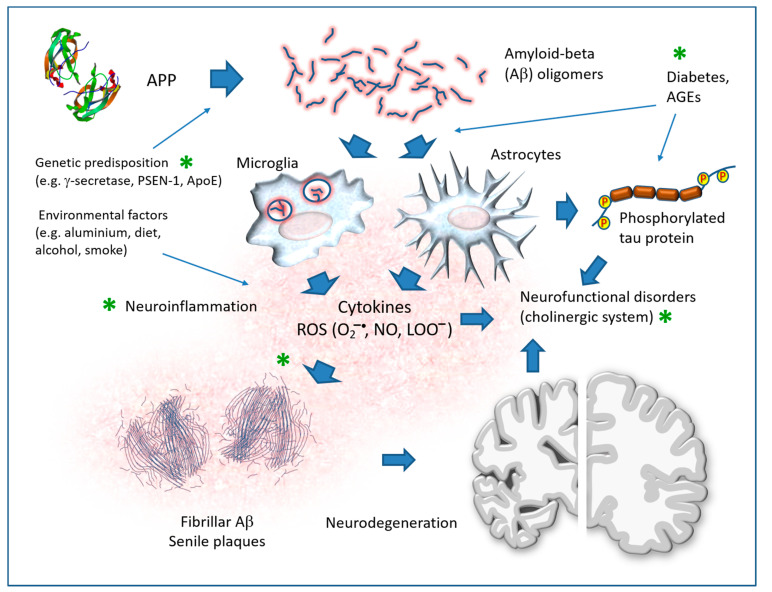
Schematic and simplified representation of AD pathology. APP (Amyloid Precursor Protein) is a transmembrane protein consisting of 770 amino acids; it is known to be the precursor of Aβ. PSEN-1: Presenilin-1; ApoE: Apolipoprotein E; AGEs: Advanced glycation end products. Asterisks indicate the proposed action points of the flavonoids, which are described in the text.

**Figure 7 antioxidants-12-00280-f007:**
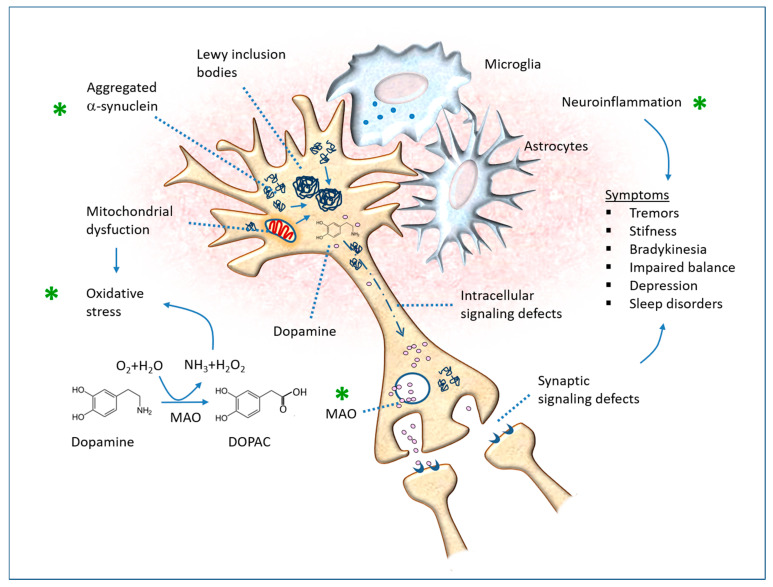
Essential pathology of PD. MAO: monoamine oxidase; DOPAC: 3,4-dihydroxyphenylacetic acid. Asterisks indicate the proposed action points of the flavonoids, which are described in the text.

**Table 1 antioxidants-12-00280-t001:** Number of publications in PubMed of the National Library of Medicine as of 10 December 2022.

Keywords in Abstracts	Keywords in Titles								
	Polyphenols	Flavonoids	Hesperidin *	Quercetin	Kaempferol	Apigenin	Naringenin	Taxifolin	Catechin
Neurodegenerative	261	145	41	146	15	28	29	2	14
Neuroinflammation	49	54	22	44	11	12	16	1	4
Depression	40	53	23	63	2	8	9	0	7
Anxiety	16	28	16	42	5	4	5	0	4
Cognitive impairment	32	20	16	42	3	6	7	1	3
Alzheimer	167	151	40	100	15	26	26	8	15
Parkinson	84	60	18	63	11	12	17	1	9
Neuroprotective	181	184	97	312	47	52	60	10	33
Antidepressant	16	28	17	29	4	8	6	0	2
Anxiolytic	2	21	8	11	8	5	2	0	0

* also includes the metabolite hesperetin.

**Table 2 antioxidants-12-00280-t002:** Models of the experimental study of the effect of hesperidin and quercetin on anxiety, depression and post-traumatic stress disorders.

Models	Treatments	Main Results
Sleep duration and sedation (mouse)	Valerian extracts containing hesperidin as well as other active ingredients	Increased sleep and sedation [136,137]
Anxiety and performance test (hole board test)	Hesperidin low doses	Synergistic effect with diazepam [138,139]
Exploration cylinder model (mouse)	*Citrus sinensis* flowers, hesperidin 11 mg/kg, i.p.	Sedative effect [140]
Forced swimming test (mouse)	*Byrsonima crassifolia* extract 500 mg/kg rich in hesperidin, rutin and quercetin	Antidepressant effect [141]
Behavioral disturbances and neurotoxicity in streptozotocin-induced diabetes (rat)	Hesperidin (50 mg/kg) once a day for 4 weeks	Antioxidant and neuroprotective effects [142]
Microarray and bioinformatic analysis of the Substantia nigra genome	*Valeriana officinalis* extract containing hesperidin and linarin	Protection from oxidative stress and neuronal hyperexcitability, hesperidin binding to the K-ATP channel regulatory subunit, the sulfonylurea receptor-1 (SUR1) [143]
Models of forced swimming and tail suspension (mouse and rat)	Hesperidin 0.1, 0.3 and 1 mg/kg, i.p.; 50–100 mg/kg, oral, according to different models	Antidepressant effect, modulation of serotonin receptors, inhibition of the L-arginine-NO pathway [144,145,146]
Hole board test and plus-maze test	Hesperidin 4–30 mg/kg i.p., 2–100 mg/kg, oral	Antidepressant and anxiolytic-like activity [147]
Depression induced by LPS (mouse)	Hesperidin 25, 50, 100 mg/kg	Antidepressant effects, cytokine decrease [148,149]
Mild brain trauma, behavioral tests (mouse)	Hesperidin 50 mg/kg for 14 days	Antidepressant effect, inhibition of cytokines and MDA [150]
Depression induced by LPS (mouse)	Orange peel extract containing hesperidin 100–200 mg/kg, i.p., 3 days pre-treatment	Significant improvements in behavioral, anorexic and biochemical parameters (oxidation and inflammation markers). Hesperidin binds NF-kappaB [151]
Models of forced swimming and suspension on the tail (mouse); PC12 cell line	Hesperidin 100, 200 mg/kg	Antidepressant effect, cytokine inhibition and BDNF stimulation [152]
Microglial cells and LPS-treated mouse	Hesperetin 5 mg/kg in vivo, 50–100 μg/mL in vitro	Decreased nitric oxide and cytokines [149,153]
Short-term stress, behavioral tests (mouse and rat)	Hesperidin 20, 50 and 100 mg/kg for 14 days	Antidepressant effect. Inhibition of MAO-A activity and decrease in tryptophan hydroxylase-1 expression in the hippocampus [146,154]
Containment stress + LPS, behavioral tests (mouse)	Hesperidin 50–100 mg/kg	Anxiolytic and antidepressant effect, reduced oxidative stress [155]
Spinal cord injury (rat)	Hesperidin 100 mg/kg	Improved motor dysfunction, decreased cytokines, increased Nrf2/ARE and antioxidant systems [156]
Depression in STZ diabetes, open field and elevated maze test (rat)	Hesperidin 50–150 mg/kg	Antidepressant and anxiolytic effects, Nrf2/ARE increase, Protein kinase A increase and other signal transduction mechanisms [157,158,159]
STZ diabetes, forced swim test (mouse)	Quercetin 50 and 100 mg/kg, i.p. for 6 weeks	Dose-dependent reduction in the period of immobility, and this effect was comparable to that of fluoxetine (5 mg/kg, i.p.) and imipramine (15 mg/kg, i.p.) [160]
Anxiety in mice exposed to the open field measured as body temperature increase	*Hypericum perforatum* and its components hipericin (0.1 mg/kg), quercitrin (0.6 mg/kg), rutin (1 mg/kg) and quercetin3-O-glucuronide (1.2 mg/kg), pre-treatment 60 min before open field exposure	Anxiolytic-like effect [161]
Acute depression from forced swimming test (rat)	Onion powder 50 mg/kg for 14 days	Reduction in immobility time, effects on dopamine metabolism [162]
Cytotoxicity of neuronal cells co-cultured with LPS-activated microglia (1 µg/mL)	Quercetin or resveratrol pre-treatment for 24 h at the dose of 0.1 µMol/L	Inhibition of the production of the inflammatory cytokines IL-1alpha and TNF-alpha by microglia. Inhibition of neuronal apoptosis in co-culture [163]
Social interaction test, forced swimming (mouse), CRH administration	Quercetin 20–40 mg/kg	Increased time in social interaction and decreased immobility time in forced swimming. Antagonism with CRH and synergism with antalarmine (CRH inhibitor) [164]
Depression after olfactory bulbectomy (rat)	Quercetin 40–80 mg/kg for 14 days	Reduction in behavioral alterations, antioxidant and anti-inflammatory effects on microglia, synergism with minocycline [165]
Depression after olfactory bulbectomy (rat)	Quercetin 25 mg/kg for 14 days	Improved behavioral testing. Normalization of lipid peroxide (LOOH) levels. Involvement of NMDA receptors and nitric oxide in the pathophysiology of depression [166]
Depression in STZ diabetes, open field and elevated maze test (rat)	Quercetin 50 mg/kg for 21 days, i.p.	Improved behavioral tests, no changes in corticosteroid levels [167]
Depression induced by 2 h of containment (mouse)	Quercetin 20 mg/kg/mL, i.p. for 15 days before the test	The treatment reversed anxiety and depression; memory performance was improved. Less lipid peroxidation and decreased acetylcholinesterase (AChE), while acetylcholine levels were increased [168]
LPS neuroinflammation (intraperitoneal injections for 1 week) (mouse)	Quercetin 30 mg/kg, i.p. for 2 weeks (1 week before LPS, 1 week after LPS)	Inhibition of glia and neuroinflammation in the cortex and hippocampus. Decrease in apoptosis. Improved memory tests [169]
Chronic Mild Unpredictable Stress (CUMS), behavioral tests such as forced swimming, tail wagging and open field movement	Quercetin 25 mg/kg after 2 weeks of stress, until 6 weeks.	Improved behavioral tests and markers of oxidative stress and 5-HT levels, decreased glutamate, TNF-alpha and IL-6 levels [43]
Psychosocial stress from intrusion (mouse)	Diet enriched with quercetin 0.5–2 g/kg long-term before stress	Improved behavioral tests, decreased astrocyte activation [170]
Zebrafish (*Danio rerio*), behavioral (colony aggregation and placement in the aquarium) and biochemical tests	Quercetin 0.01, 0.1, 1, 10, 100 and 1000 μg/L in the test aquarium	Quercetin at lower concentrations exerted beneficial effects by reducing inflammatory cytokines and oxidative stress. Conversely, when quercetin reached 1000 μg/L, it exerted harmful effects [171]
Chronic Mild Unpredictable Stress (CUMS) (mouse)	Quercetin 10, 20, 40 mg/kg/day for 3 weeks	Dose-dependent antidepressant effect (20 and 40 mg). Improved antioxidant indices and Nrf2 in the hippocampus [172]
Chronic Mild Unpredictable Stress (CUMS) (mouse)	Quercetin 50 mg/kg/day for 8 weeks	Normalization of metal and trace element levels in serum, reduction in oxidative stress [173]
LPS-induced depression (mouse)	Quercitrin single dose 10 mg/kg, i.p.	Antidepressant effect, increase in neuroplasticity signaling molecules, reduction in IL-10, IL-1beta and TNF-alpha in serum, as well as PI3K/AKT/NF-kappaB and MEK/ERK pathways in the hippocampus [174]
Cultured microglia cells exposed to LPS (100 ng/mL) for 24 or 48 h	Pre-treatment with quercetin 30–100 μMol/L for 1 h	Reduction in inflammatory cytokines, iNOS and the NLRP3 inflammasome; beneficial effects on mouse models as well [175]
Head injury (rat)	Quercetin 5, 20 or 50 mg/kg i.p. at 0.5, 12 and 24 h after trauma	Anti-edema effect, attenuation of cortical inflammatory responses and activation of the Nrf2/HO-1 pathway in the cortex [176]
Methamphetamine-induced anxiety (mouse)	Quercetin 50 mg/kg/day for 3 weeks	Antipsychotic, antioxidant and mitochondrial protection activity; reduction in cytokine production by cultured astrocytes [177]
Psychosocial stress from intrusion (mouse)	Quercetin 25, 50 and 100 mg/kg, i.p. and ginseng 50 mg/kg, i.p. for 14 days	Improvement of anxiety and depression tests, normalization of the neuroendocrine axis, enhancement of BDNF and inhibition of neuroinflammation [178]
LPS neuroinflammation (single injection of 1 mg/kg) in zebrafish, behavioral tests	Quercetin 50–100 mg/kg, i.p. for 7 days	Anxiolytic-like effect, reduction in TNF-alpha and IL-1beta, Lipoperoxides, nitrites and AChE and increase in GSH [179]

**Table 3 antioxidants-12-00280-t003:** Experimental studies of the effect of hesperidin and quercetin on neurotoxicity models.

Models	Treatments	Main Results
Ischemia by carotid artery occlusion for 30 min followed by reperfusion (rat)	Pre-treatment with hesperidin 50–100 mg kg for 7 days	Improved neurobehavioral alterations, attenuated oxidative damage, restored the activities of antioxidant and mitochondrial enzymes in the brain [204]
CCl4 neurotoxicity (rat)	Hesperidin 200 mg/kg for 8 days	Cytoprotective effects, decrease in LPO (lipoperoxides), increase in glutathione peroxidase [205]
Cainic acid excitotoxicity (rat)	Hesperidin 10–50 mg/kg, i.p.	Protection of hippocampal cells and decrease in extracellular glutamate [41]
Cisplatin neurotoxicity (rat)	Hesperidin 50 mg/kg	Reduction in cellular damage, decrease in lipoperoxidation [206]
AlCl3-induced neurotoxicity, memory and maze test (mouse)	Hesperidin 50–100 mg/kg	Reduction in oxidative stress and cognitive impairment [207]
Cadmium neurotoxicity (rat)	Hesperidin 40 mg/kg for 21 days	Reduction in changes in oxidative stress biomarkers such as lipoperoxides, AChE, monoamine oxidase, ATPase [208,209]
Cognitive impairment from L-methionine-induced hyperhomocysteinemia (rat), maze test.	Hesperidin 100 mg/kg	Improved behavioral testing, abrogation of oxidative stress [210]
Lysolecithin demyelination, visual impairment (rats)	Hesperetin 20 mg/kg for 14 or 21 days	Improvement of visual potentials, reduction in microglia, increase in myelin basic protein [211,212]
Transient cerebral ischemia induced with middle cerebral artery occlusion (mouse)	Yuzu (a citrus fruit) extract and its active ingredient hesperidin 10 mg/kg	Protection of the blood–brain barrier and claudin-5, reduction in MMP-3/9 (a type of protease) [213]
Fluoride toxicity, behavioral and cognition tests (rat)	Hesperidin 100–200 mg/kg for 8 weeks	Neurobehavioral improvement and restoration of brain biochemical changes (AChE and antioxidant activity). Reduction in inflammatory cytokines [32,214]
Methotrexate toxicity, biochemical assays (Rat)	Hesperidin 100 mg/kg	Improved levels of BDNF and Nrf2 in the hippocampus and prefrontal cortex [215]
Sodium arsenite (NaAsO2)-induced toxicity, biochemical assays (rat)	Hesperidin 100–200 mg/kg for 2 weeks	Antioxidant, anti-inflammatory and antiapoptotic effect [216]
Toxicity from emamectin benzoate insecticide, neurobehavioral and cognitive tests (rat)	Hesperidin 100 mg/kg for 8 weeks	Improvement of neural functions and reduction in oxidative stress and inflammation [217]
Neurotoxicity from the subcutaneous injection of D-galactose (mouse)	Quercetin 5–10 mg/kg for 8 weeks	Anxiolytic-like effect, increased activity in the open field and learning ability and memory. Increase in SOD, decrease in MDA [218]
Cadmium chloride neurotoxicity (rat)	Quercetin 15 mg/kg i.p. for 30 days	Reduction in MDA levels and increase in enzyme antioxidants in the frontal cortex tissue. Reduction in caspase immunoreactivity [219]
Aluminum neurotoxicity (rat)	Daily pre-treatment with quercetin 10 mg/kg, intragastric administration	Reduced ROS production, increased SOD activity, Bcl-2 upregulation, structural protection of mitochondria [220]
Neurotoxicity from the subcutaneous injection of D-galactose (mouse)	Quercetin 20 or 50 mg/kg for 8 weeks.	Improved memory and behavioral tests, increased expression of Nrf2, HO-1 and SOD [221]
Aluminum chloride neurotoxicity (rat)	Quercetin 50 mg/kg and/or alpha-lipoic acid 20 mg/kg i.p. for 2 weeks	Inhibition of lipid peroxidation and normalization of antioxidant enzymes and acetylcholine esterase activity in the brain; synergy with alpha-lipoic acid [222]
Zidovudine (azidothymidine, AZT)-induced neurotoxicity and neuroinflammation (mouse)	Quercetin 50 mg/kg for 8 days	Inhibition of inflammatory cytokines produced by astrocytes, mediated by the inhibition of Wnt5a (Wnt Family Member 5A, implicated in chronic inflammatory disorders) [223]
Vincristine peripheral neurotoxicity (rat)	Quercetin 25 and 50 mg/kg for 12 days	Increased Nrf2 and HO-1 activities in sciatic nerve tissue, decreased neuronal apoptosis [224]
Iron oxide particle neurotoxicity (rat)	Quercetin 25, 50 and 100 mg/kg for 30 days	Dose-dependent decrease in MDA and brain tissue lesions related to neuronal apoptosis [225]
Silver nanoparticles (AgNPs)-induced oxidative neurotoxicity (rat)	Quercetin 50 mg/kg for 30 days	Counteracts the pathological effects of AgNPs through the modulation of tight junction proteins, Nrf2 and paraoxonases and the modulation of pro-inflammatory cytokines [226]

**Table 4 antioxidants-12-00280-t004:** Studies on the effects of hesperidin and quercetin in experimental models of AD.

Models	Treatments	Main Results
APP/PS1 transgenic mouse	Hesperidin 100 mg/kg a day, 16 weeks	Improvement of cognitive and motor functions, reduction in biochemical disorders [257]
APP/PS1 transgenic mouse	Hesperidin 100 mg/kg for 10 days	Improvement of social interactions, anti-inflammatory effects and a reduction in Aβ peptides in the mouse hippocampus and cortex [258]
Cognitive impairment from the intracerebroventricular injection of STZ (rat)	Hesperidin 100–200 mg/kg for 15 days	Improved memory, modulation of acetylcholine esterase activity. Improved GSH and the inhibition of NF-kappaB, inducible nitric oxide synthase, cyclooxygenase-2 [259]
AD-type lesions after the administration of aluminum (AlCl3) (rat)	Hesperidin 100 mg/kg for 60 days	Attenuation of hippocampal lesions and behavioral disturbances by reducing Aβ(1–40) levels and by inhibiting β- and γ-secretase. Attenuation of acetylcholine esterase [260,261].
Scopolamine-induced AD-type cognitive and memory impairment (mouse)	Hesperidin 100–200 mg/kg for 10 days	Potentiate the therapeutic effect of donepezil (a drug used to counteract the symptoms of dementia) [262]
APP/PS1 transgenic mouse	Hesperidin 40 mg/kg for 90 days	Attenuation of cognitive impairment, increase in Nrf2/ARE and antioxidant enzymes, deactivation of the receptor of the advanced glycation end products (RAGE)-mediated pathway [263]
AD-like cognitive and memory impairment from the intracerebroventricular injection of STZ (rat)	Hesperetin and nano-hesperetin 10, 20 mg/kg for 3 weeks	Improvement in symptoms, increased activity of antioxidant enzymes and GSH levels and decreased malondialdehyde in the hippocampus [264]
AD-type lesions after the administration of aluminum (AlCl3) (rat)	Hesperidin 100 mg/kg for 60 days	Behavioral improvements, reduction in the phosphorylation of the tau protein, markers of inflammation, nitric oxide and apoptosis [71]
Aβ(1–42) injection-induced brain degeneration (mouse)	Hesperetin 50 mg/kg for 6 weeks	Attenuated oxidative stress and inflammation, as assessed by Nrf2, ROS and NF-kappaB expression in the hippocampus, cortex and HT22 cells in vitro [70]
Scopolamine-induced AD-type cognitive and memory impairment (mouse)	Hesperetin 1, 5 or 50 mg/kg for 3 days	Improved non-spatial and spatial learning and reduced memory impairment [265]
AD-type lesions after the administration of aluminum (AlCl3) (rat)	Hesperidin 125–250 mg/kg for 2 weeks	Protection against cognitive impairment, reduction in acetylcholine esterase and Aβ levels [266]
Scopolamine-induced AD-type cognitive and memory impairment (rat)	Hesperidin 100 mg/kg for 28 days	Reduction in spatial memory deficits, redox imbalance, Aβ(1–42) and AChE [267]
APP/PS1 transgenic mouse	Quercetin 50 mg/kg for 4 months	Increased BDNF levels and decreased Aβ in the hippocampus, cognitive improvement [268]
Scopolamine-induced amnesia (zebrafish)	Pre-treatment with quercetin or rutin 50 mg/kg, i.p.	Protection in memory and behavioral tests [269]
Amnesia induced by the intracerebral injection of Aβ(25–35) (mouse)	Quercetin 10–40 mg/kg for 8 days after Aβ injection	Improved learning and memory abilities, decreased neurovascular oxidation, improved cholinergic activity, inactivation of the RAGE-mediated pathway [270]
Motor paralysis in the human Aβ-producing transgenic strain of the Caenorhabditis elegans nematode	Quercetin doses >10 µMol/L for 48 h.	Dose-dependent decrease in aggregated Aβ and related paralysis [271]
APP/PS1 transgenic mouse	Quercetin 50 mg/kg for 16 weeks	Reduction in learning deficits, reduction in scattered senile plaques and improvement of mitochondrial dysfunction [272]
Memory impairment induced by cadmium exposure (rat)	Quercetin 5, 25 or 50 mg/kg for 45 days	Protection of AChE and Na(+),K(+)-ATPase, reduction in oxidative stress markers [273]
Elderly triple transgenic AD model mice	Quercetin 25 mg/kg i.p. every 48 h for 3 months	Decreased beta-amyloidosis, tauopathy, astrogliosis and microgliosis in the hippocampus and amygdala. Decreased BACE1-mediated APP cleavage. Improved performance in learning and memory [274]
APP23 transgenic mouse (human amyloid precursor protein overexpression)	Long-term diet with 20% casein and 0.5% quercetin	Reduction in presenilin 1 expression and Aβ secretion, prevention of mental deterioration [275]
Motor paralysis in the human Aβ-producing transgenic strain of the Caenorhabditis elegans nematode	Quercetin 33 µMol/L for 48 h.	Reduction in symptoms of paralysis, stimulation of metallothionein and lengthening of the lifespan [276]
STZ-induced memory impairment (rat)	Quercetin 80 mg/kg i.p., for 3 weeks, plus regular exercise for 60 days	Quercetin and exercise synergistically improved spatial memory and reduced oxidative stress [277]
AD induced by aluminum chloride (AlCl3) for 28 days.	Quercetin 25–50 mg/kg for 28 days after induction	Attenuation of behavioral deficits, reduction in insoluble Aβ plaques in the hippocampus, increased activity of metalloproteases [278]
5XFAD transgenic mouse expressing human APP and PSEN1 with a total of five AD-linked mutations	Diet enriched with quercitrin 50–100 mg/kg for 3 months	Improved behavioral testing and reduction in amyloid plaques, inhibition of microglial cytokines [279]

**Table 5 antioxidants-12-00280-t005:** Studies on the effects of hesperidin and quercetin in experimental models of PD.

Models	Treatments	Main Results
Cortical neurons treated with 5-S-cysteinyl-dopamine in vitro	Various polyphenols including hesperetin and quercetin 0.1–3.0 μMol/L	Protection from neuronal damage [316,317]
Rotenone-induced apoptosis in human neuroblastoma SK-N-SH cells in vitro	Hesperidin 4–30 nMol/L	Preservation of mitochondrial function, inhibition of ROS and the molecular mechanisms of apoptosis (Bax, cytochrome c and caspases 3 and 9, and the downregulation of Bcl-2) [318].
6-Hydroxydopamine (6-OHDA)-induced PD (elderly mouse)	Hesperidin 50 mg/kg for 28 days	Protection of glutathione peroxidase and catalase activity, striatum dopamine levels, improvement of cognitive symptoms and modulation of proinflammatory cytokines [319,320,321]
PD induced by iron (Fe) intake in Drosophila melanogaster (fruit fly)	Diet with hesperidin 10 µMol/L	Decreased Fe concentration in the head, normalization of dopamine levels and cholinergic activity and improvement of motor function impaired by Fe; inhibition of oxidative stress and mitochondrial protection [322]
Model of PD on 6-OHDA-intoxicated SH-SY5Y neuroblastoma cells	Hesperidin 1–10 µMol/L	Preservation of cell vitality, rebalancing of Ca^2+^ homeostasis, reduction in oxidative stress [323,324]
PD and AD induced in Drosophila melanogaster with the UAS-GAL4 system	Hesperidin 1, 5 or 10 mMol/L in food throughout life	Protection from the development of the disease and symptoms [325]
Iron dextran-induced PD (mouse)	Hesperidin or Coumarin 50–100 mg/kg and Desferal 25 mg/kg 4 times/week for 4 weeks	Hesperidin chelates iron, like Desferal [312]
PD induced by 6-OHDA (rat)	Quercetin 30 mg/kg for 14 days	Increased striatal dopamine, antioxidant enzymes and neuronal survival [326]
Neurotoxicity due to manganese excess (500 μMol/L Mn for 24 h) on cultured SK-N-MC neurons	Quercetin 10–20 μg/mL	Protective effect of quercetin on cell mortality, decreased NF-kappaB but increased heme oxygenase-1 (HO-1) and Nrf2 [327]
Neurotoxicity due to manganese excess (MnCl2 15 mg/kg i.p. for 8 days) (rat)	Quercetin 25–50 mg/kg for 8 days	Decreased oxidative stress, decreased various apoptotic markers and upregulated anti-apoptotic Bcl-2 proteins and SOD [327]
MitoPark mouse model with the inactivation of mitochondrial transcription factor A in dopaminergic neurons	Quercetin 25 mg/kg for 6 weeks	Counteraction of behavioral deficits, striatal dopamine depletion and neuronal cell loss. Activation of two cell survival kinases and BDNF, increased mitochondrial bioenergetic capacity [328]
PD induced by 6-OHDA (rat)	Quercetin 10 and 25 mg/kg (in normal form or in nanocrystals) for 4 weeks	Memory preservation, increase in antioxidant enzymes and total glutathione. Higher efficacy in the form of nanocrystals [329]
Parkinsonism induced by the administration of rotenone at a dose of 1.5 mg/kg for 28 days (rat)	Quercetin, 15–50 mg/kg, co-administered with rotenone	Dose-dependent neuroprotective effects by improving behavioral tests and signs of oxidative stress. Drug targets could be aromatic L-amino acid decarboxylase and catechol-O-methyltransferase [330].
Parkinsonism induced by the administration of rotenone at a dose of 1.5 mg/kg in rats for 8 days (rat)	Quercetin, 50 mg/kg for 2 weeks before and after rotenone	Restoration of motor and cognitive functions, improvement of antioxidant enzymes in the brain [331]
PD induced by 6-OHDA in vitro on PC12 cells and in vivo (rat).	Quercetin 20 μMol/L in vitro and 30 mg/kg for 14 days in vivo	Improved mitochondrial function, reduced oxidative stress, increased levels of PINK1 and Parkin markers and decreased α-synuclein protein expression in PC12 cells. Reduction in PD-like behaviors and α-synuclein accumulation in rats [332]
Parkinsonism induced by the multiple-dose administration of rotenone at a dose of 1.5 mg/kg (rat)	Quercetin 5–20 mg/kg subcutaneously for 3 days	Attenuation of neurobehavioral deficits caused by rotenone. Attenuation of striatal redox stress and neurochemical dysfunction of the NF-kappaB and IkappaKB genes [333]

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
