# Peer review of "Neuroprotective Potentials of Flavonoids: Experimental Studies and Mechanisms of Action"

_antioxidants, 2023, doi:10.3390/antiox12020280_

Round 1

Reviewer 1 Report

The manuscript adequately compiles the available bibliography on the protective effect of the flavonoids hespereridin and quercetin in the development of different neurological pathologies. The review is systematic and correctly structured, accompanied by explanatory tables and figures.

Although the author justifies, through the bibliographic search, why focus on quercetin and hespereridin, since the title is more generic, existing knowledge on the neuroprotective effects of another large family of flavonoids such as flavan-3-ols should have been collected, especially (epi)catechin.

On the other hand, the number of citations, despite being a review article, seems very high. It would be more useful for readers if the author carried out a greater selection work, citing only the most relevant ones in each of the items.

Some minor comments:

-          Should try that in table 1 the names of the flavonoids enter in one line

-          Please, check superscripts and radicals in lines 169-170, 223…

Author Response

  1. The manuscript adequately compiles the available bibliography on the protective effect of the flavonoids hespereridin and quercetin in the development of different neurological pathologies. The review is systematic and correctly structured, accompanied by explanatory tables and figures.

R: I thank the Honorable Reviewer for his/her positive comment

  1. Although the author justifies, through the bibliographic search, why focus on quercetin and hesperidin, since the title is more generic, existing knowledge on the neuroprotective effects of another large family of flavonoids such as flavan-3-ols should have been collected, especially (epi)catechin.

R: I agree that e(epi) catechin should be mentionated. Thanks to this note of the Reviewer, I have added the bibliographic search of catechin (which includes also epicatechin) to Table 1 and reported this flavan-3-ol on the Figure 1.

  1. On the other hand, the number of citations, despite being a review article, seems very high. It would be more useful for readers if the author carried out a greater selection work, citing only the most relevant ones in each of the items.

R: I did my best and cut 14 non essential citations. Actually, I believe that a merit of a review like this is that it cites the largest number of publications, and it would be difficult to choose some and exclude others, even out of fairness to researchers in this field. In any case, the completeness of the citations is found only in the tables, while in the text the topics have been condensed in a way that is easier to read and only the most important citations are mentioned.

Some minor comments:

  1. Should try that in table 1 the names of the flavonoids enter in one line

 R: Thanks to this advice, in Table 1 I have written the names of the flavonoids vertically so that they fit in one line

  1. Please, check superscripts and radicals in lines 169-170, 223…

R: Thanks to this right note of the Reviewer, I have corrected superscripts and radicals of molecules O2- ,OH (hydroxyl radical), NO3- (peroxynitrite) at the indicated pages

Reviewer 2 Report

This review article is based on the huge and commendable knowledge of the author, and actually it encompasses a vast range of references. I believe this kind of review is only possible when it is by a true professional. I cannot help admiring the author for his/her academic achievement as a professional. Although the review is already semi-perfect, let me say just a few comments.

#As the author states in the very beginning of the abstract (Neurological and neurodegenerative diseases, particularly those related to aging), if we think about neuroprotection, the aging of the brain is an important issue. For example, although it is true that AD has a purely disease-induced pathogenesis, the base of its pathogenesis is the aging of the brain. The pathogenesis of AD is, so to speak, composed of the aging of the brain and a disease-induced pathogenesis, the latter of which is overlying the former. But for the aging of the brain, I believe, AD would not exist (off course, except for some familial cases); if human beings died, say, around 30s or 40s, as in the past, they would not be affected with AD. The same scenario is true for PD. Thinking this way, the aging of the brain is an interesting and important issue in terms of its association with neuroprotective potentials of flavonoids. With this in mind, I read the paper. Chapters 1-4 deal with introduction and chemical features of flavonoids; this is good. While on the other hand, chapter 5, anxiety, etc; 6, neurotoxicity; 7, AD; 8, PD; then, how about inserting another chapter that specifically deals with the issue of the aging of the brain (or the body in general) and flavonoids? Are there some studies specifically devoted to this issue? Are there any reports revealing potential rejuvenating or anti-aging effects of flavonoids? If a chapter specifically addressing this issue were available, it would be of great help for the understanding, say, of chapters 7, AD or 8, PD. This is just a remark of mine, but a soft suggestion for the author. Whether the paper is accordingly revised or not is up to the author. After all, the paper is, already in the present form, is quite satisfactory.

 #Just a bit of grammatical or stylish questions; please proofread the Ms again.

 -page 2, line 80 or page 27, line 994: Alzheimer disease > Alzheimers disease (just like the others)

-page 4, line 147: amyloid Aβ oligomers > amyloidbeta (Aβ) oligomers (in Figure 6 as well)

-page 6, line 205: trigger > triggers

-page 21, line 792: LPS which cause a > LPS, which causes a

-page 28, line 1034: products, which lead to > products, which leads to

 -etc.

Author Response

  1. This review article is based on the huge and commendable knowledge of the author, and actually it encompasses a vast range of references. I believe this kind of review is only possible when it is by a true professional. I cannot help admiring the author for his/her academic achievement as a professional. Although the review is already semi-perfect, let me say just a few comments.

R: I thank the Honorable Reviewer for his/her positive comment

  1. #As the author states in the very beginning of the abstract (‘Neurological and neurodegenerative diseases, particularly those related to aging’), if we think about neuroprotection, the ‘aging of the brain’ is an important issue. For example, although it is true that AD has a purely disease-induced pathogenesis, the base of its pathogenesis is the aging of the brain. The pathogenesis of AD is, so to speak, composed of the aging of the brain and a disease-induced pathogenesis, the latter of which is overlying the former. But for the aging of the brain, I believe, AD would not exist (off course, except for some familial cases); if human beings died, say, around 30s or 40s, as in the past, they would not be affected with AD. The same scenario is true for PD. Thinking this way, ‘the aging of the brain’ is an interesting and important issue in terms of its association with neuroprotective potentials of flavonoids. With this in mind, I read the paper. Chapters 1-4 deal with introduction and chemical features of flavonoids; this is good. While on the other hand, chapter 5, anxiety, etc; 6, neurotoxicity; 7, AD; 8, PD; then, how about inserting another chapter that specifically deals with the issue of ‘the aging of the brain (or the body in general)’ and ‘flavonoids’? Are there some studies specifically devoted to this issue? Are there any reports revealing potential rejuvenating or anti-aging effects of flavonoids? If a chapter specifically addressing this issue were available, it would be of great help for the understanding, say, of chapters 7, AD or 8, PD. This is just a remark of mine, but a soft suggestion for the author. Whether the paper is accordingly revised or not is up to the author. After all, the paper is, already in the present form, is quite satisfactory.

R: It is not easy to distinguish the "pure" anti-aging effects from the effects of preventing diseases related to old age. In any case, thanks to this interesting reviewer's note, I have added a small chapter (n. 9) paragraph on the potential anti-aging effects of flavonoids, with a brief bibliography to support it.: “Flavonoids have positive effects on aging-related diseases {Zhao, 2022 #23593}, but few studies have explored whether they also exert independent antiaging effects {Fan, 2022 #23601}. On simple and unicellular models, senolytic effects of flavonoids have been observed which prolong the biological life of senescent cells {Malavolta, 2016 #23615}, of the yeast S. cerevisiae {Zahoor, 2022 #23594}, of human keratinocytes {Stanisic, 2020 #23629}{Gu, 2022 #23600}, adipocytes {Zoico, 2021 #23624}{Islam, 2023 #23621}, fibroblasts {Chondrogianni, 2010 #23619}, umbilical vein endothelial cells {Fan, 2022 #23627} and intervertebral disc (nucleus pulpy){Lewinska, 2022 #23625}. A few experiments have observed an extension of the life span of the nematode Caenorhabditis elegans {Tao, 2021 #23603}{Niu, 2022 #23597}{Wang, 2020 #23604}, of Drosophila melanogaster {Zhang, 2022 #23622} and of mouse { Guo, 2022 #23599}. In the latter case, the effect was accredited to the action on the intestinal microbiota. Although the antioxidant effects of flavonoids make the biological plausibility of an anti-aging effect very high, further studies are needed to clarify whether they really have positive effects in prolonging human life.”

  1. #Just a bit of grammatical or stylish questions; please proofread the Ms again.

             -page 2, line 80 or page 27, line 994: Alzheimer disease > Alzheimer’s disease (just like the others)

-page 4, line 147: amyloid Aβ oligomers > amyloid–beta (Aβ) oligomers (in Figure 6 as well)

-page 6, line 205: trigger > triggers

-page 21, line 792: LPS which cause a > LPS, which causes a

-page 28, line 1034: products, which lead to > products, which leads to

R: OK these changes have been made (including Figure 6)  and the manuscript has been carefully proofread
